 

**HydroMix v1.0: a new Bayesian mixing framework for attributing uncertain hydrological**
**sources**
Harsh Beria[1], Joshua R. Larsen[2], Anthony Michelon[1], Natalie C. Ceperley[1], Bettina Schaefli[1]
[1] Institute of Earth Surface Dynamics, University of Lausanne, Lausanne, Switzerland
[2] School of Geography, Earth and Environmental Sciences, University of Birmingham, United
Kingdom
**Abstract**
Tracers have been used for over half a century in hydrology to quantify water sources with
the help of mixing models. In this paper, we build on classic Bayesian methods to quantify
uncertainty in mixing ratios. Such methods infer the probability density function (pdf) of the
mixing ratios by formulating pdfs for the source and target concentrations and inferring the
underlying mixing ratios via Monte Carlo sampling. However, collected hydrological samples
are rarely abundant enough to robustly fit a pdf to the sources. Our approach, called
HydroMix, solves the linear mixing problem in a Bayesian inference framework where the
likelihood is formulated for the error between observed and modelled target variables, which
corresponds to the parameter inference set-up commonly used in hydrological models. To
address small sample sizes, every combination of source samples is mixed with every target
tracer concentration. Using a series of synthetic case studies, we evaluate the performance of
HydroMix. We then use HydroMix to show that snowmelt accounts for 60-62% of
groundwater recharge in a Swiss Alpine catchment (Vallon de Nant), despite snowfall only
accounting for 40-45% of the annual precipitation. Using this example, we then demonstrate
the flexibility of this approach to account for uncertainties in source characterization due to
different hydrological processes. We also address an important bias in mixing models that
arises when there is a large divergence between the number of collected source samples and
their flux magnitudes. HydroMix can account for this bias by using composite likelihood
functions that effectively weights the relative magnitude of source fluxes. The primary
application target of this framework is hydrology, but it is by no means limited to this field.
Keywords: stable water isotopes; hydrograph separation; isotopic lapse rate; rain; snow;



# 1 Introduction

Most water resources are a mixture of different water sources that have travelled via distinct flow paths in the landscape (e.g. streams, lakes, groundwater). A key challenge in hydrology is to infer source contributions to understand the flow paths to a given water body using a source attribution technique. A classic example is the two-component hydrograph separation model to quantify the proportion of groundwater and rainfall in streamflow, often referred to as "pre-event" water vs "event" water (Brewer et al., 2011; Burns et al., 2001; Buttle et al., 1995; Dusek and Vogel, 2018; Joerin et al., 2002; Klaus and McDonnell, 2013; Lopes et al., 2018; Schmieder et al., 2016; W. et al., 2009). Other examples include estimating the proportional contribution of rainfall and snowmelt to groundwater recharge (Beria et al., 2018; Earman et al., 2006; Jasechko et al., 2014, 2017; Jasechko and Taylor, 2015; Jeelani et al., 2010; Kohfahl et al., 2008; O'Driscoll et al., 2005; Rose, 2003; Winograd et al., 1998), fog to the amount of throughfall (Scholl et al., 2011, 2002; Uehara and Kume, 2012), and soil moisture (at varying depths) and groundwater to vegetation water use (Dawson and Ehleringer, 1991; Ehleringer and Dawson, 1992; Evaristo et al., 2016, 2017; Evaristo and McDonnell, 2017; Rothfuss and Javaux, 2017; Vargas et al., 2017; Zhao et al., 2016).

The primary goal of such attribution in hydrology is to infer the contribution of different sources to a target water body, where the tracer can be an observable compound like a dye, or a conservative solute, or even a proxy for chemical composition such as electrical conductivity. The key requirement is that the concentration of the tracer is distinguishable between different sources. The sources are then assumed to linearly mix in the target water body as follows:

$$\rho_1 S_1^k + \rho_2 S_2^k + \cdots + \rho_n S_n^k = Y^k. \qquad 1$$

where $Y^k$ is the concentration of the $k^{th}$ tracer in the target mixture, $S_i^k$ is the concentration of the $k^{th}$ tracer in source $i$. $\rho_i$ ($i=1, .., n$) are the fractions of all sources in the mixture, with $\sum_{i=1}^n \rho_i = 1$. The system of the $n$ linear equations can be solved if the number of tracers is $n$-1, leading to a system of $n$ equations and $n$ unknown variables.

The stable isotope composition of hydrogen and oxygen in water (subsequently referred to as 'stable isotopes of water') are used as tracers in hydrology. Other commonly used tracers include electrical conductivity (Hoeg et al., 2000; Laudon and Slaymaker, 1997; Lopes et al., 2018; Pellerin et al., 2007; Weijs et al., 2013) and conservative geochemical solutes such as chloride and silica (Rice and Hornberger, 1998; Wels et al., 1991).

Classically, Eq. (1) is solved by assigning an average tracer concentration to each source, estimated typically from time or space-averages of observed field data (Maule et al., 1994; Winograd et al., 1998). Alternatively, Bayesian mixing models can be used, which explicitly acknowledge the variability of source tracer concentrations estimated from observed samples (Barbeta and Peñuelas, 2017; Blake et al., 2018). Rather than a single estimate of source contributions, Bayesian approaches yield full probability density functions (pdfs) of the fraction of different sources in the target mixture (Parnell et al., 2010; Stock et al., 2018), hereafter referred to as 'mixing ratios'.





Bayesian mixing was first developed in ecology to estimate the proportion of different food
sources to animal diets (Parnell et al., 2010; Stock et al., 2018). Hydrological applications of
such models are still rare (Blake et al., 2018; Evaristo et al., 2016, 2017; Oerter et al., 2019). In
a Bayesian mixing model, a statistical distribution is fitted to both the measured source tracer
concentrations, and to the measured tracer concentrations from the target (e.g. river,
groundwater, vegetation). The distribution of the mixing ratios is then inferred via Bayesian
inference as follows: Random samples are drawn from the source distributions and from a
given prior distribution of the mixing ratios. Based on these samples, the target tracer
concentrations are calculated according to Eq. (1). The likelihood of a given (drawn) set of
mixing ratios is calculated by comparing the modelled target tracer concentrations with their
observed values. With recent advances in probabilistic programming languages like Stan
(Carpenter et al., 2017), this has become a relatively simple task.
However, the key limitation with the above approach is that the source compositions are
assumed to come from standard statistical distributions. Typically, the sources are assumed
to be drawn from Gaussian distributions, which can be fully characterized by the mean and
variance of the data available for each source (Stock et al., 2018). This limits both the potential
applicability and the insight that can be gained from tracer information in hydrology because:
1) The mean and variance may not accurately reflect the statistical properties of the
21       source composition.
2) If there is a large amount of information on the source composition, the mean and
23       variance may be an unnecessary simplification of its variability.
3) If the source compositions have a low number of samples, then the mean and variance
25       estimates may be poorly constrained.

An additional complication in hydrology comes from the fact that observed point-scale
samples do not necessarily capture the tracer concentrations in the actual sources, which are
spatially distributed and whose contribution can be temporally variable depending on the
state of the catchment (Harman, 2015). For instance, if we were to characterize the
contribution of snowmelt to groundwater, we need to capture (1) the temporal evolution of
the isotopic ratio of snowmelt, which strongly varies in space (Beria et al., 2018; Earman et al.,
2006), and (2) the temporal evolution of the area actually covered by snow. This spatially and
temporally distributed nature of the sources can be hard to account for in both the analytical
and the Bayesian mixing approaches.

To overcome the above limitations, we present a new mixing approach for hydrological
applications, called HydroMix. This approach does not require a parametric description of
observed source or target tracer concentrations. Instead, HydroMix formulates the linear
mixing problem in a Bayesian inference framework similar to hydrological rainfall-runoff
models (Kavetski et al., 2006a), where the mixing ratios of the different sources are treated as
model parameters. Thereby, HydroMix explicitly uses the whole dataset of observed source
tracer concentrations instead of reducing it to its few statistical moments. An advantage of
this approach is that additional model parameters can be incorporated in the framework to
describe how the source tracer concentrations might be modified according to specific
hydrologic processes that can be decided and explored by the user.



In this paper, we first describe the theoretical details of HydroMix for a simple case study with
two sources, one mixture and one tracer (Section 2). Section 3 presents synthetic and real-
world case studies that demonstrate the accuracy, robustness and flexibility of HydroMix. In
the synthetic case study, we use a conceptual hydrologic model to simulate tracer
concentrations. We also introduce a composite likelihood function that accounts for the
magnitude of the different sources. The results of these applications are presented in Section
4 before summarizing the main outcomes, applicability, and limitations of HydroMix in Section
8  5.

## 2  Model description and implementation
This section details the general modeling approach underlying HydroMix for a system with
two sources and one tracer. This is followed by a detailed discussion on the choice of the
parameter inference approach used.
### 2.1  Linear mixing model with non-concomitant observed data
For a system with two sources that combine linearly to form a mixture, the mixing model can
be formulated as:
$$\rho S_1(t) + (1 - \rho)S_2(t) = Y(t + \tau), \qquad\qquad 2$$
where $S_1(t)$ is the concentration of tracer in source 1 at timestep $t$, $S_2(t)$ is the concentration
of tracer in source 2 at timestep $t$, $Y(t + \tau)$ is the concentration of the mixture (i.e. the tracer
concentration in the target) at timestep $t + \tau$, $\rho$ is the mixing ratio and $\tau$ is the time delay
between the time when source enters the system and the time when it is observed in the
mixture. As an example, for a case where the two sources are snowmelt and rainfall and the
mixture is groundwater, $\rho$ represents the proportional groundwater recharged from
snowmelt and $\tau$ represents the average time lag for rain and snowmelt to reach the
groundwater once they enter into the soil.
The two parameters in this system, the mixing ratio ($\rho$) and the time delay ($\tau$), can be inferred
via classical Bayesian parameter inference which is widely used in hydrology (Kavetski et al.,
2006a, 2006b; Schaefli and Kavetski, 2017). This implies taking an observed timeseries of the
target (e.g. the tracer concentration in groundwater) and building a vector of model residuals:
$$\varepsilon_t = \tilde{Y}_t - \hat{Y}_t, \qquad\qquad 3$$
where $\tilde{Y}_t$ represents the observed mixture concentration and $\hat{Y}_t$ represents the simulated
mixture concentration. However, in real environmental systems like that of groundwater
recharge from rainfall and snowmelt, there are four major difficulties which can prevent the
inference of $\rho$ and $\tau$ from the observed data.
i. $\rho$ and $\tau$ strongly vary in time depending on catchment conditions such as soil moisture
(Benettin et al., 2017; Harman, 2015).
ii. Long time series of the tracer concentration in both the sources and mixture are rare.
iii. The effect of seasonality in precipitation can make the inference of $\tau$ very difficult in case the
goal is to understand the intra-annual recharge dynamics.





iv. The tracer concentration in the different sources are generally measured at point scales
whereas the tracer concentration in the target integrates inputs over the entire source area.
Section 3.5 introduces such an example and proposes a solution in these cases.
Our practical solution to limitation ii) is to assume that tracer concentrations in the two
sources are functions of observable point processes:
$$S_i(t) = f_i\big(P_i(t)\big), \qquad\qquad 4$$
where the function $f_i$ represents the transformation from the point to the catchment scale for
source $i$. Limitation iii) can be relaxed by assuming a long enough timestep (eg: long term
groundwater recharge dynamics) or a short enough timestep (eg: event based hydrograph
separation (Klaus and McDonnell, 2013)) such that we can neglect $\tau$ and write Eq. (2) as:
$$\rho S_1'(\Delta t) + (1 - \rho)S_2'(\Delta t) = Y'(\Delta t), \qquad\qquad 5$$
where the $'$ signifies time-integrated processes. The space- and time-integrated processes $S_i$
are not directly observable. We thus need to make the simplifying assumption that any
observed point-scale tracer concentration $p_i$ in a given source $i$ (e.g., the isotopic ratio of
snowmelt) represents a sample of the space- and time-integrated processes $S_i$. This
assumption is in fact implicitly underlying most of the existing hydrological mixing models
where point samples are used to characterize a spatial process and where the time reference
of the samples is discarded.
By utilizing all the available measurements $\{p_1'\}_{i=1..n}$ and $\{p_2'\}_{j=1..m}$ of the two sources in the
above model, with $n$ samples of source 1 and $m$ samples of source 2, we can build
$n \times m$ predictions and compare them with the $q$ observed samples of the target as:
$$\varepsilon_{ij}^k = \tilde{Y}_{obs}^k - \hat{Y}_{ij}, \qquad\qquad 6$$
where $\tilde{Y}_{obs}^k$ is the $k$-th observed target concentration out of a total number of $q$ target
concentrations.
Assuming that the residuals can be described with a Gaussian error model with a mean of zero
and constant variance $\sigma$,
$$\varepsilon \sim N(0, \sigma), \qquad\qquad 7$$
we can compute the likelihood function of the residuals as the joint probability of all the
residuals:
$$L_j(\tilde{Y}_{obs}|S_1, S_2, \boldsymbol{\theta}) = \prod_{k=1}^{q} \prod_{j=1}^{m} \prod_{i=1}^{n} (2\pi\sigma^2)^{-0.5} \exp\left(-\frac{1}{2}\frac{(\tilde{Y}_{obs}^k - \hat{Y}_{ij})^2}{\sigma^2}\right), \qquad 8$$
where $\boldsymbol{\theta}$ represents all the model parameters. The above Gaussian error model could in
principle be replaced with any other stochastic process. However, the Gaussian error model





has been shown to be relatively robust in this kind of an application (Schaefli and Kavetski,
2    2017).
In the case of linear mixing between two sources, the two model parameters considered at
this stage are the mixing ratio $\rho$ and the error variance $\sigma$. The error variance can either be
computed from the observed residuals or be treated as a model parameter (Kuczera and
Parent, 1998; Schaefli et al., 2007). For the examples shown in this paper, the error variance
is computed from the residuals.
In order to avoid numerical problems, we use the log-likelihood form of Eq. (8):
$$\log L_j(\tilde{Y}_{obs}|S_1, S_2, \theta) = \sum_{k=1}^{q} \sum_{j=1}^{m} \sum_{i=1}^{n} -0.5 \left[ 2\pi\sigma^2 + \frac{\left(\tilde{Y}_{obs}^k - \hat{Y}_{ij}\right)^2}{\sigma^2} \right].$$    9
2.2    Parameter inference in a Bayesian framework
Following the general Bayes' equation, the posterior distribution of the model parameters
can be written as:
$$p(\boldsymbol{\theta}|S_1, S_2, \tilde{Y}) = \frac{p(\tilde{Y}|\theta, S_1, S_2)p(\theta)}{p(\tilde{Y}|S_1, S_2)},$$    10
where $p(\boldsymbol{\theta})$ is the prior distribution of the model parameters and $p(\tilde{Y}|\boldsymbol{\theta}, S_1, S_2)$ is the
likelihood function. The denominator of Eq. (10) can generally not be computed as that would
require integration over the whole parameter space which is computationally expensive,
which is why Eq. (10) is reduced to:
$$p(\boldsymbol{\theta}|S_1, S_2, \tilde{Y}) \propto p(\tilde{Y}|\boldsymbol{\theta}, S_1, S_2)p(\boldsymbol{\theta}).$$    11
Two methods are traditionally used in hydrology to infer the posterior distribution from Eq.
(11), Markov Chain Monte Carlo (MCMC) sampling and importance sampling. In the case of
MCMC sampling, a common approach is the Metropolis algorithm (Kuczera and Parent, 1998;
Schaefli et al., 2007; Vrugt et al., 2003). This method is suited when the overall computational
cost for inferring the posterior distribution is high, either because of a large number of model
parameters or a computationally intensive model. In importance sampling, the posterior
distribution is obtained from weighted samples drawn from the so-called importance
distribution. For typical multivariate hydrological problems, the only possible choices for the
importance distribution are either uniform sampling over a hypercube or sampling from an
over-dispersed multi-normal distribution (Kuczera and Parent, 1998). A stochastic process is
defined as over-dispersed when the variance of the underlying distribution is greater than its
mean (Inouye et al., 2017). The sampling distributions in such cases have large variance,
allowing sufficient sampling over the entire parameter range.

Given the low number of model parameters in HydroMix, we infer the posterior distribution
by random Monte Carlo sampling. The prior distribution of the mixing ratio is assumed to be
uniform between 0 and 1. With the uniform prior assumption, the posterior distribution is



dependent only on the likelihood function. In order to compute the posterior distribution, only model runs with the highest likelihood score, corresponding to the top 5 percentile of the model runs, are retained. This also highlights the key difference between MCMC and importance sampling.

The prior distribution of additional model parameters (if applicable) are discussed in the corresponding case study section. The error model variance is not jointly inferred with other model parameters but calculated for each sample parameter set from the residuals according to Eq. (6).

## 3    Case studies

We provide a comprehensive overview of the performance of HydroMix based on a set of synthetic case studies (case studies 3.1 and 3.2) and a real-world application to demonstrate the practical relevance for hydrologic applications (case studies 3.4 and 3.5). The first case study demonstrates the ability of HydroMix to converge on the correct posterior distribution for synthetically generated data. The second case study uses a synthetic dataset of rain, snow and groundwater isotopic ratios using a conceptual hydrologic model, and compares the results of HydroMix to the actual mixing ratios assumed to generate the data set. It then weights the sources samples and evaluates the effect of weighting on the mixing ratio. In the third and fourth case studies, HydroMix is applied to observed tracer data from an Alpine catchment in the Swiss Alps to infer source mixing ratios and an additional parameter (isotopic lapse rate).

### 3.1   Mixing using Gaussian distributions

In this example, sources $S_1$ and $S_2$ are drawn from two Gaussian distributions with different means ($\mu_1, \mu_2$) and standard deviations ($\sigma_1, \sigma_2$) and combined to form the mixture $Y$ with a constant mixing ratio $\rho$:

$$\rho S_1 + (1 - \rho)S_2 = Y. \hspace{2cm} 12$$

Assuming the two distributions are independent, the resultant mixture is normally distributed with mean and variance defined as:

$$Y \sim N(\rho\mu_1 + (1 - \rho)\mu_2, \ \rho^2\sigma_1^2 + (1 - \rho)^2\sigma_2^2). \hspace{2cm} 13$$

A given number of samples are drawn from the distributions of $S_1$ and $S_2$ and of the mixture $Y$. The posterior distribution of the mixing ratio, $p(\boldsymbol{\rho}|\widetilde{S_1}, \widetilde{S_2}, \widetilde{Y})$, is then inferred using HydroMix for i) a case where the two source distributions are well identifiable, and ii) a case where the distributions have a large overlap. Different values of mixing ratios are tested, with ratios varying from 0.05 to 0.95 in steps of 0.05.

The sensitivity of HydroMix to the number of samples drawn from $S_1$, $S_2$ and $Y$, along with the time to convergence is assessed based on the sum of the absolute error between the estimated mixing ratio $\hat{\rho}$ and its true value $\rho$.



## 3.2   Mixing with a time series generated using a hydrologic model
In this case study, we build a conceptual hydrologic model where groundwater is assumed to
be recharged directly from rainfall and snowmelt. Stable isotopes of water in deuterium ($\delta^2$H)
is used to see how the isotopic ratio in groundwater evolves under different assumptions of
rain and snow recharge efficiencies.
Synthetic time series are generated for precipitation, isotopic ratio in precipitation and air
temperature at a daily timestep. For generating the precipitation time series, the time
between two successive precipitation events is assumed to be a Poisson process with the
precipitation intensity following an exponential distribution (Botter et al., 2007; Rodriguez-
Iturbe et al., 1999). Time series of air temperature and of isotopic ratios in precipitation are
obtained by generating an uncorrelated Gaussian process with the mean following a sine
function (to emulate a seasonal signal) and with constant variance (Allen et al., 2018; Parton
and Logan, 1981). The separation of precipitation into rainfall ($P_r$) and snowfall ($P_s$) is done
based on a temperature threshold approach (Harpold et al., 2017), where the fraction of
rainfall $f_r(t)$ at time step t is computed as a function of air temperature $T(t)$:
$$f_r(t) = \begin{cases} 0 & \text{if } T(t) < T_L \\ \frac{T(t)-T_L}{T_H-T_L} & \text{if } T_L \leq T(t) \leq T_H \\ 1 & T(t) > T_H, \end{cases}$$    14
where $T_L$ and $T_H$ are the lower and upper threshold bounds. In this case study, $T_L$ and $T_H$ are
set to -1 °C and +1 °C. The evolution of the snow water equivalent (SWE) in the snowpack ($h_s$)
is computed as:
$$\frac{dh_s(t)}{dt} = P_s(t) - M_s(t),$$    15
where $M_s$ is the magnitude of snowmelt, computed using a degree-day approach as proposed
by Schaefli et al., (2014):

$$M_s = \begin{cases} a_s(T(t) - T_m), & \text{if } T(t) > T_m \\ 0 & \text{otherwise} \end{cases},$$    16

where $a_s$ is the degree-day factor (set here to 2.5 mm/°C/day) and $T_m$ is the threshold
temperature at which snow starts to melt (set to 0 °C). The snowpack is assumed to be fully
mixed, and the isotopic ratio of snowpack is computed as:

$$\frac{d(h_s(t)C_s(t))}{dt} = C_p(t)P_s(t) - C_s(t)M_s(t),$$    17

where $C_s$ is the isotopic ratio of snowpack and $C_p$ is the isotopic ratio of precipitation. The
amount of groundwater recharge ($R$) is the sum of groundwater recharged from rainfall and
snowmelt:



$$R(t) = R_r P_r(t) + R_s M_s(t), \tag{18}$$

where $R_r$ and $R_s$ are the rainfall and snowmelt recharge efficiencies. Recharge efficiency is defined as the fraction of rainfall or snowmelt that reaches groundwater and is assumed to be a constant value. The groundwater storage is assumed to be fully mixed, and the isotopic ratio of groundwater is computed as:

$$\frac{d(G(t)C_g(t))}{dt} = R_r C_p(t) P_r(t) + R_s C_s(t) M_s(t) - C_g(t)Q(t), \tag{19}$$

where $C_g$ is the isotopic ratio in groundwater, $G$ is the volume of groundwater and $Q$ is the amount of groundwater outflow to the stream defined as:

$$Q(t) = k(G(t) - G_C), \tag{20}$$

where $k$ is the recession coefficient and $G_C$ is a constant groundwater storage that does not interact with the stream (added here to avoid zero flow). This formulation follows the linear groundwater reservoir assumption used in numerous hydrological modeling frameworks (Beven, 2011). The volume of the groundwater storage is computed as:

$$\frac{dG(t)}{dt} = R(t) - Q(t). \tag{21}$$

The model is run for a period of 100 years, allowing the system to reach a long term steady state. Only the last 2 years of the model runs are used to obtain the time series of isotopic ratios in rainfall, snowmelt and groundwater. These years are then used to estimate the mixing ratio of snowmelt in groundwater, which is the fraction of groundwater recharged from snowmelt. Rainfall and snowmelt samples are the two sources and groundwater samples represent the mixture. For the HydroMix application, all the rainfall and snowmelt samples are used, whereas for groundwater, only one isotopic ratio per month is used (randomly sampled). The mixing ratios inferred using HydroMix are compared to the actual recharge ratio obtained from the hydrologic model as:

$$R_s^a = \frac{\sum_t R_s M_s(t)}{\sum_t R(t)}, \tag{22}$$

where $R_s^a$ represents the proportion of groundwater recharge derived from snowmelt, summed over all the time steps. The numerical implementation of the evolution of isotopic ratio in snowpack and groundwater are given in the Appendix.

### 3.3    Weighting mixing ratios in the hydrologic model

In Section 3.2, rainfall and snowmelt samples are not weighted by the magnitude of their fluxes while computing the mixing ratios with HydroMix. As all rainfall and snowmelt samples are used, the weights are implicitly determined by the number of rainfall and snowmelt events, instead of their magnitudes. This is a general problem in all mixing approaches and has not been adequately acknowledged in the literature. Ignoring the weights may lead to biased mixing estimates if the proportional contribution of one of the components (e.g.:





rainfall or snowmelt) is low, but the number of samples obtained to represent that component
is proportionally much higher (Varin et al., 2011). For example, in a given catchment, the
amount of total snowfall maybe a small proportion of the annual precipitation, but the
number of days when snowmelt occurs maybe comparable to the total number of rainfall days
in a year. If this is not specified *a priori*, HydroMix may overestimate the proportion of
groundwater being recharged from snowmelt. To account for this, we introduce a weighting
factor in the likelihood function originally formulated in Eq. (8), to make a new composite
likelihood (Varin et al., 2011):
$$L_j(\breve{Y}_{obs}|S_1, S_2, \boldsymbol{\theta}) = \prod_{k=1}^{q} \prod_{j=1}^{m} \prod_{i=1}^{n} \left[ (2\pi\sigma^2)^{-0.5} \exp(-\frac{1}{2} \frac{(\breve{Y}_{obs}^k - \hat{Y}_{ij})^2}{\sigma^2}) \right]^{w_i w_j},$$ 23
where $i$ and $j$ correspond to snowmelt and rainfall samples, and the weights $w_i$ and $w_j$ reflect
the proportion of snowmelt and rainfall contributing to groundwater recharge (Vasdekis et
al., 2014), where $w_i$ is expressed as:
$$w_i = \frac{R_i S_i}{\sum_{i=1}^{n} R_i S_i},$$ 24
where $R_i$ is the magnitude and $S_i$ is the isotopic ratio of the $i^{th}$ snowmelt event. Rain weights
($w_j$) are also expressed similarly to Eq. (24). The obtained mixing ratio estimates are then
compared with the unweighted estimates (in Section 3.2) to see if weighting by magnitude
makes a significant difference.
3.4   Real case study: Snow ratio in groundwater in Vallon de Nant
The objective of this case study is to infer the proportional contributions of snow versus
rainfall to the groundwater of an Alpine headwater catchment, Vallon de Nant (Switzerland),
using stable water isotopes.
3.4.1 Catchment description

Vallon de Nant is a 13.4km² catchment located in the Vaud Alps in South-West of Switzerland
(Figure 1), with elevation ranging from 1253 m to 3051 m asl. Steep slopes form a major part
of the catchment with a mean catchment slope of around 36° (Thornton et al., 2018). At lower
elevations, a dense forest dominated by *Picea abies* covers 14% of the catchment area. At
around 1500 m asl., there is an active pasture area with scattered trees and an open forest
dominated by *Larix decidua*. Additional species scattered throughout the catchment include
*Pinus sp., Alnus sp.* and *Acer pseudoplatanus*. Alpine meadows cover most of the higher
elevation land surfaces. Despite the relatively low elevation, there is a small glacier with an
extended moraine that cover 4.4% and 10.1% of the catchment area. A large part (28% of
catchment area) of the hillslopes are composed of steep rock walls. At lower to mid-
elevations, talus slopes account for about 6% of the catchment area.

Vallon de Nant has a typical Alpine climate, with around 1900 mm of annual precipitation and
a mean air temperature of 1.8 °C (Michelon, 2017). For this paper, long term climate statistics
are computed using MeteoSwiss gridded precipitation and air temperature dataset from



1961-2015 (Isotta et al., 2013; MeteoSwiss, 2016, 2017). Applying a simple temperature
threshold (0 and 1 °C) to observed precipitation indicates that on average, 40-45% of the total
precipitation falls as snow in the catchment. There is a small degree of seasonality in
precipitation, with higher precipitation between June to August, and lower precipitation in the
months of September and October.

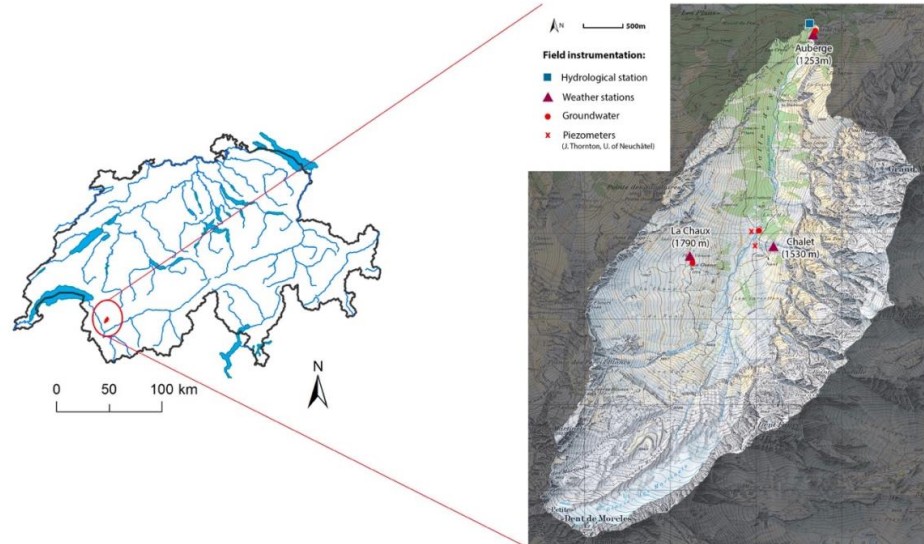

**Figure 1.** Map showing Vallon de Nant along with the locations of meteorologic and hydrologic
observations and the frequent sampling sites. Composite samples of precipitation were
collected at the weather stations. Groundwater samples were collected at the groundwater
monitoring points and the installed piezometers. The groundwater piezometers were installed
by James Thornton from University of Neuchâtel (Thornton et al., 2018).
3.4.2 Data collection
Vallon de Nant has been extensively monitored since February 2016. Water samples are
collected from streamflow, rain, snowpacks and groundwater at different elevations, which
are then analyzed for the isotopic ratios in deuterium ($\delta^2$H) and oxygen-18 ($\delta^{18}$O). Due to
logistical constraints, snowmelt water is not collected, thus snowpack samples are used as a
proxy for snowmelt. A summary of the isotopic data is shown in Table 1.
**Table 1.** Summary of the isotopic data ($\delta^2$H and $\delta^{18}$O) collected in Vallon de Nant between
February 2016 to July 2017

| Sample name | Number of samples | Lowest elevation | Highest elevation |
|---|---|---|---|
| Rainfall | 32 | 1253 | 1773 |
| Top snowpack layer | 80 | 1241 | 2455 |
| Groundwater | 22 | 1253 | 1779 |





3.4.3 Model implementation
HydroMix is used to estimate the proportion of snow recharging groundwater (subsequently
referred to as 'snow recharge coefficient'). In order to obtain a pdf of the snow recharge
coefficient, isotopic ratios in all the water samples from rain, snowpack and groundwater are
used. A uniform prior distribution is assigned to the snow recharge coefficient, which varies
between 0 and 1, representing the entire range of possible values. Groundwater isotopic ratio
is estimated using Eq. (12).
3.5   Introduction of an additional model parameter
In any mixing analysis, it may be useful or desirable for users to specify an additional model
parameter that is able to modify the tracer concentrations based on their process
understanding of the system. In the case of Alpine catchments with large elevation gradients,
stable isotopes in precipitation often exhibit a systematic trend with elevation, becoming
more depleted in heavier isotopes with increasing elevation. This is also known as the 'isotopic
lapse rate' (Beria et al., 2018). In typical field campaigns, because of logistical challenges,
precipitation samples are collected only at a few points in a catchment, with often fewer
precipitation samples at high elevations. This leads to oversampling at lower elevations, and
under sampling at higher elevations, which can bias mixing estimates. This has been found
specially relevant for hydrograph separation in forested catchments (Cayuela et al., 2019). To
allow a process compensation for this, an additional lapse rate factor is introduced in which
each observed point scale sample (observed at a given elevation) is corrected to a reference
elevation as follows:
$$\bar{r} = \frac{\sum_{j=1}^{k}[\alpha(e_j - e) + r]a_j}{\sum_{j=1}^{k} a_j},$$     25
where $r$ is the isotopic ratio in precipitation collected at elevation $e$, $\bar{r}$ is the catchment
averaged isotopic ratio in precipitation, $\alpha$ is the isotopic lapse rate factor, and $e_j$ is the
elevation of the $j$-th elevation band where the catchment is divided into $k$ elevation bands.
These bands are obtained by constructing a hypsometric curve of the catchment (Strahler,
32   1952).


The lapse rate factor is allowed to modify both rainfall and snowpack isotopic ratios to obtain
a catchment averaged isotopic ratio, which is then used in the mixing model. Using this
formulation of an isotopic lapse rate makes the following implicit assumptions: (1)
precipitation storms on aggregate move from the lower part of the catchment to the upper
part of the catchment thus creating a lapse rate effect, and (2) precipitation falls uniformly
over the catchment. It is important to note that the isotopic lapse rate is different from the
precipitation lapse rate, i.e., the rate of change of precipitation with elevation is different from
the rate of change of precipitation isotopic ratio with elevation.

The prior distribution of the isotopic lapse rate is specified based on isotopic data collected
across Switzerland under the Global Network of Isotopes in Precipitation (GNIP) program
(IAEA/WMO, 2018). Using the monthly isotopic values collected in between 1966 and 2014,





average lapse rate values are obtained for both $\delta^2H$ and $\delta^{18}O$. These were (-)1.94 ‰/100m for
$\delta^2H$, and (-)0.27 ‰/100m for $\delta^{18}O$ (Beria et al., 2018).
A uniform prior distribution is assigned to the isotopic lapse rate parameter, with the lower
bound specified as three times the Swiss lapse rate for both $\delta^2H$ and $\delta^{18}O$. The observed
isotopic lapse rate data from Switzerland suggests average lapse rates are weakly negative;
however, positive lapse rates can *a priori* not be excluded for the case study catchment.
Accordingly, we do not specify an upper lapse rate bound of zero but set it as three times the
Swiss lapse rate (Table 2). In the case of Vallon de Nant, the elevation ranges from 1253 m to
3051 m asl. For computing the Swiss lapse rate, the elevation range over which the monthly
precipitation samples were collected was 300 m to 2000 m asl.
**Table 2.** Prior distribution of the different model parameters as specified to HydroMix

| Variable | Prior distribution | Lower bound | Upper bound |
|---|---|---|---|
| Snow recharge coefficient | Uniform | 0 | 1 |
| Isotopic lapse rate in $\delta^2H$ | Uniform | (-)5.82 ‰/100m | (+)5.82 ‰/100m |
| Isotopic lapse rate in $\delta^{18}O$ | Uniform | (-)0.81 ‰/100m | (+)0.81 ‰/100m |

## 15   4   Results

The results for the different case studies are discussed in the sections below.
### 4.1   Mixing with normal distributions
The mean and standard deviations used to generate the low and high variance source
distributions for the synthetic case studies are summarized in Table 3. We randomly generated
40 samples from each of the two source distributions and from the target distribution, and
varied the mixing ratios between 0.05 and 0.95 in 0.05 increments. However, it should be
noted that HydroMix permits using different number of samples for the sources and the
mixture.
For the low variance case, the posterior distributions obtained with HydroMix with 1000
Monte Carlo (MC) simulations reproduce closely the theoretical mean of the mixing ratios
used to generate the synthetic data (Figure 2). However, for the high variance case, the
posterior distributions do not capture the true underlying mixing ratios. This is partly due to
the poor identifiability of the sources (given that their distributions are highly overlapping),
and partly due to the small sample size of 40. Interestingly, the model performance improves
slightly with an increasing number of samples (Figure 3a) but markedly with an increase in the
number of MC runs (Figure 3b). The performance is measured here in terms of the absolute
error between the posterior mixing ratio mean and the true mean, summed over all tested
ratios from 0.05 to 0.95
**Table 3.** Mean and variance of the two sources $S_1$ and $S_2$ drawn from Normal distribution

| Dataset | $S_1$ mean (standard deviation) | $S_2$ mean (standard deviation) |
|---|---|---|
| Low variance | 10 (0.5) | 20 (0.5) |





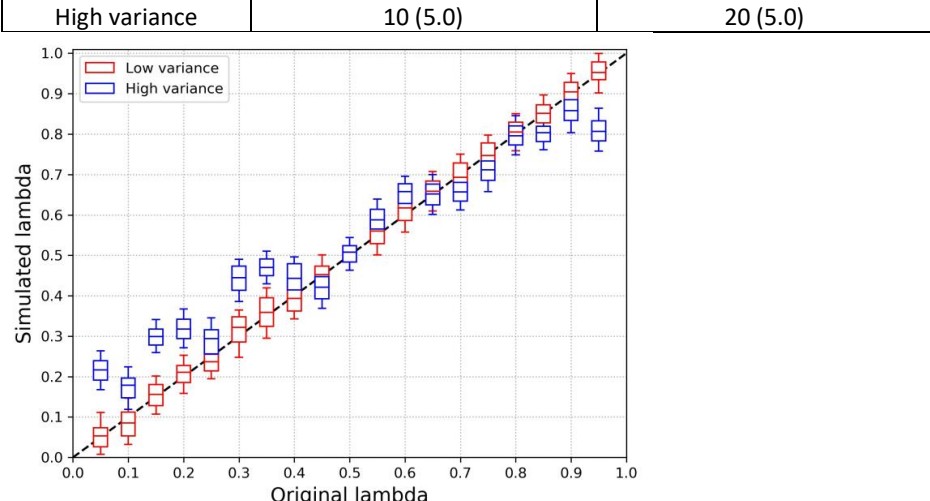

**Figure 2.** Scatterplot showing boxplots of the mixing ratio ($\rho$) values inferred using HydroMix
for the low and high variance synthetic case of Table 3. The boxplots shows the median value,
with the box extending from 25th to 75th percentile values. The number of Monte Carlo runs
is 1000 and the boxplot represents the top 10 percentile values of the mixing ratio.

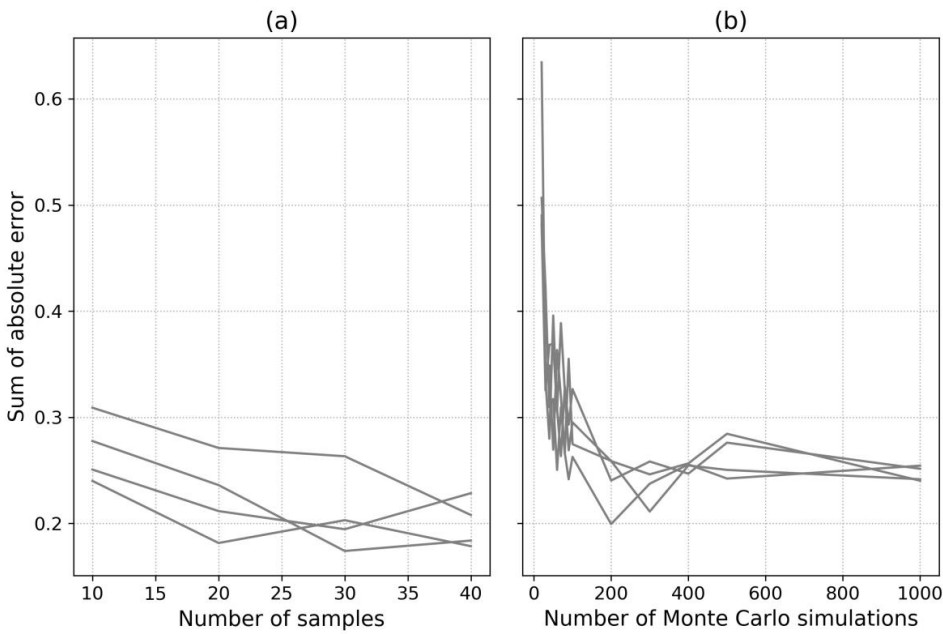

**Figure 3.** Performance of HydroMix in terms of the absolute error between the posterior
mixing ratio mean and the true mean, summed over all tested ratios plotted as a function of
(a) number of samples drawn for the two sources (100 Monte Carlo simulations) and (b)
number of MC simulations for sample size 10. The four lines in the plot correspond to four
different random seeds that were used to initialize HydroMix. The underlying dataset used is
the low variance dataset shown in Table 3.



4.2   Contribution of rain and snow to groundwater recharge using a hydrologic model
The parameters used to generate daily precipitation, air temperature and precipitation
isotopic ratios for a run time of 100 years are shown in Table 4. The static volume of
groundwater that does not interact directly with the stream, $G_C$ is set to 1000 mm. Figure 4
shows the resulting variation in the isotopic ratio of groundwater over the entire 100 year
period, showing the system achieves a steady state condition after ~15 years of simulation.
**Table 4.** Parameters used to generate time series of precipitation, air temperature and
isotopic ratios in precipitation. $\mu$ represents the mean, $A$ is the amplitude and $\phi$ the time lag
of the underlying sine function. For the precipitation process, $\mu$ is the mean intensity on days
with precipitation. The resulting mean winter length (air temp. below 0°C) is 119.5 days.

| Variable | Parameter values |
|---|---|
| Precipitation | # events/year = 30, $\mu$ = 33.45 mm/day |
| Air temperature | $\mu$ = 4 °C, $A$ = 8 °C, $\phi$ = -π/2 |
| Precipitation isotopic ratio | $\mu$ = (-80) ‰, $A$ = 40 ‰, $\phi$ = -π/2 |

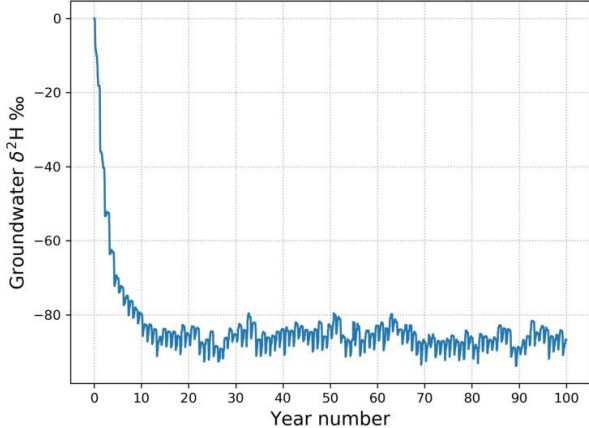

**Figure 4.** Evolution of the modeled isotopic ratio in groundwater over a 100-year period with
$R_r$= 0.3 and $R_s$=0.6.
The mixing ratio is estimated with HydroMix using: (1) samples of the isotopic ratio in snowfall,
and (2) samples of the isotopic ratio in snowmelt. The two sample distributions differ, as
shown in Figure 5, where the variability of the isotopic ratio is lower in snowmelt when
compared to snowfall. In the model at hand, this reduction is obtained because of mixing
occurring within the snowpack, leading to homogenization, thus reducing the variability in the
isotopic ratio of snowmelt. In field data, such a reduction in variability is also generally
observed (Beria et al., 2018), as a result of the homogenization as modelled here and from
more complex snow physical processes, which lie beyond the scope of this study.





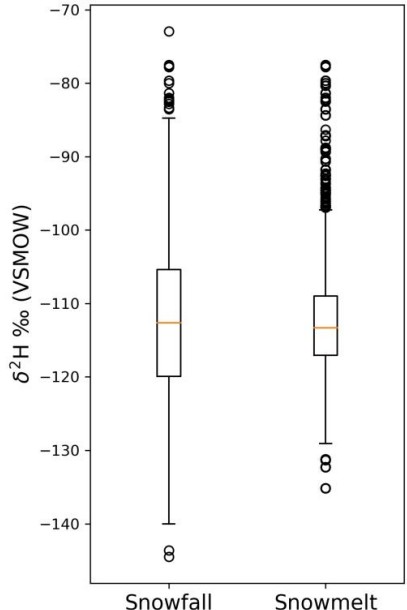

**Figure 5.** Boxplot showing the variability in the isotopic ratio of snowfall and snowmelt as
simulated by the hydrologic model. The boxplot extends from 25th to 75th percentile value,
with the median value depicted by the orange line. The whiskers extend up to 1.5 times of the
interquartile range. The black circles are the outliers.
The mixing ratios inferred with HydroMix are very similar regardless of whether snowfall or
snowmelt is used across the entire range of recharge efficiencies (Figure 6). This provides
confidence in the use of snowfall samples as a proxy for snowmelt when estimating mixing
ratios. However, it is clear from Figure 6 that an important bias emerges between the
estimated mixing ratio from HydroMix and the actual mixing ratio known from the hydrologic
model, especially for high and low mixing ratios.
This bias can be expected to emerge where the source contributions are not weighted
according to their fluxes, which to our knowledge has not been explicitly addressed in the
hydrological literature. As already discussed in Section 3.3, the absence of sample weighting
typically induces a bias when there is a large divergence between the amount of samples taken
over a certain period (e.g. one year) to characterize a source, and the magnitude of source
flux over that period (e.g. 40 snow and 10 rain samples taken to characterize the two sources,
where snow only accounts for a very small portion (e.g. 10%) of the annual precipitation).





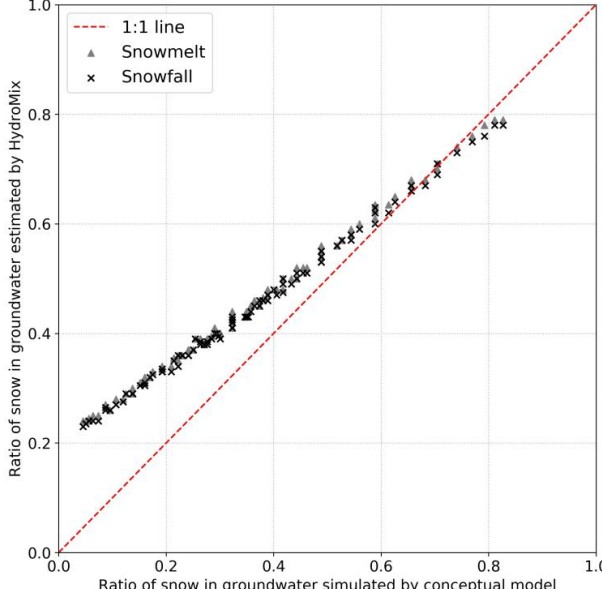

**Figure 6.** Ratios of snow in groundwater estimated with HydroMix plotted against ratios obtained from the hydrologic model for the last two years of simulation. Also shown are the separate results obtained by using samples of either snowmelt or snowfall. The full range of ratios is obtained by varying rainfall and snowmelt recharge efficiencies from 0.05 to 0.95. The number of rainfall, snowfall and snowmelt days are 39, 24 and 107 in the last two years of simulation.

4.3    Effect of weights on estimates of mixing ratios using a hydrologic model

After taking into account the magnitude of rainfall and snowmelt events in the composite likelihood function of Eq. (23), it is clear that much of the un-weighted biases can be removed (Figure 7). The most significant improvement is seen at very low mixing ratios where the divergence between the conceptual model and the mixing model estimates error reduces by almost 50%. In this study, we have used a relatively simple normalization based weighting function (Eq. (24)). Testing other weighting functions which have been proposed in the past (Vasdekis et al., 2014) is certainly possible, and is left for future research.



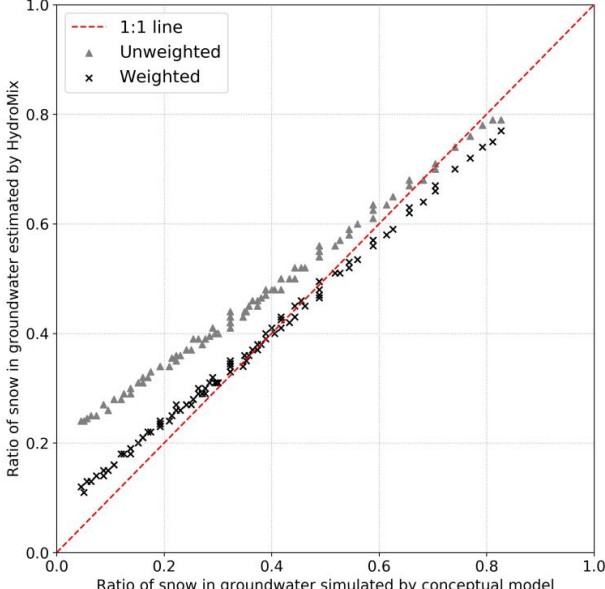

2
**Figure 7.** Ratios of snow in groundwater estimated using HydroMix plotted against ratios
obtained from the hydrologic model, for both weighted and unweighted mixing scenarios. The
full range of ratios is obtained by varying rainfall and snowmelt recharge efficiencies from 0.05
to 0.95. The number of rainfall, snowfall and snowmelt days are 39, 24 and 107 in the last two
years of simulation.

4.4   Inferring fraction of snow recharging groundwater in a small Alpine catchment
along with an additional model parameter
Using the dataset from an Alpine catchment (Vallon de Nant, Switzerland), HydroMix
estimates that 60-62% of the groundwater is recharged from snowmelt (using unweighted
approach), with the full posterior distributions shown in Figure 8a. This estimate is consistent
for both the isotopic tracers ($\delta^2$H and $\delta^{18}$O), which are often used interchangeably in the
hydrologic literature (Gat, 1996). Comparing this recharge estimate to the proportion of total
precipitation that falls as snow (around 40-45%, see Section 3.4.1), suggests that snowmelt is
more effective at reaching the aquifer than an equivalent amount of rainfall falling at a
different period of the year. Similar results have been obtained in a number of previous
studies across the temperate and mountainous regions of the world (see Table 1 in the work
of Beria et al., (2018) for a summary).



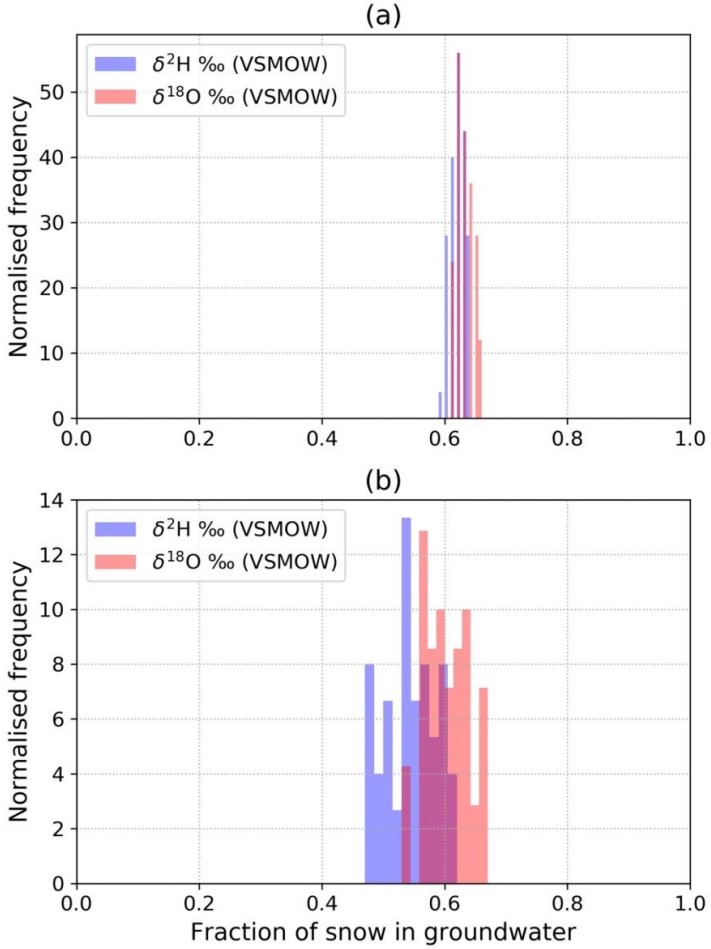

**Figure 8.** Histogram showing the fraction of snow recharging groundwater in Vallon de Nant using the isotopic ratios in $\delta^2$H and $\delta^{18}$O (a) without correcting for lapse rate and (b) after correcting for lapse rate.

As can be seen from Figure 8a, the estimated distribution of snow ratio in groundwater is very narrow. This can be explained by the fact that we assume that the collected precipitation samples represent the variability actually occurring in the catchment. To overcome this limitation, we infer an additional parameter called the isotopic lapse rate that accounts for the spatial heterogeneity in terms of catchment elevation. As shown in Figure 9, the posterior distributions of the isotopic lapse rate (for both $\delta^2$H and $\delta^{18}$O) largely overlap with the spatially averaged isotopic lapse rate as estimated from precipitation isotopes across Switzerland. The overlap with the average Swiss isotope lapse rate suggests our inferred lapse rates are reasonable, with the spread in the estimates likely reflecting the temporal variation in the catchment specific isotope lapse rate that can develop from a wide range of moderating factors (e.g. air masses contributing precipitation without traversing the full elevation range of the catchment due to varying trajectories). The Swiss lapse rate is constructed as a long





term spatial average, whereas the inferred isotopic lapse rate in Vallon de Nant is constructed
from the temporal variations in the isotopic ratios. This makes the comparison more
informative than definitive. In any case, these results demonstrate that it is relatively
straightforward to jointly infer multiple parameters within the HydroMix modeling framework
provided users have a mechanistic basis for their interpretation.
However, an important consequence of additional parameter inference without providing
additional data or constraints is an increase in the degree of freedom, which can then increase
the uncertainty on source contributions. This effect is seen in Figure 8b, especially in contrast
with the previous result in Figure 8a, where the median mixing ratios of the posterior
distributions remain similar (~0.6), but the spread increase drastically, from 0.005 to 0.2.

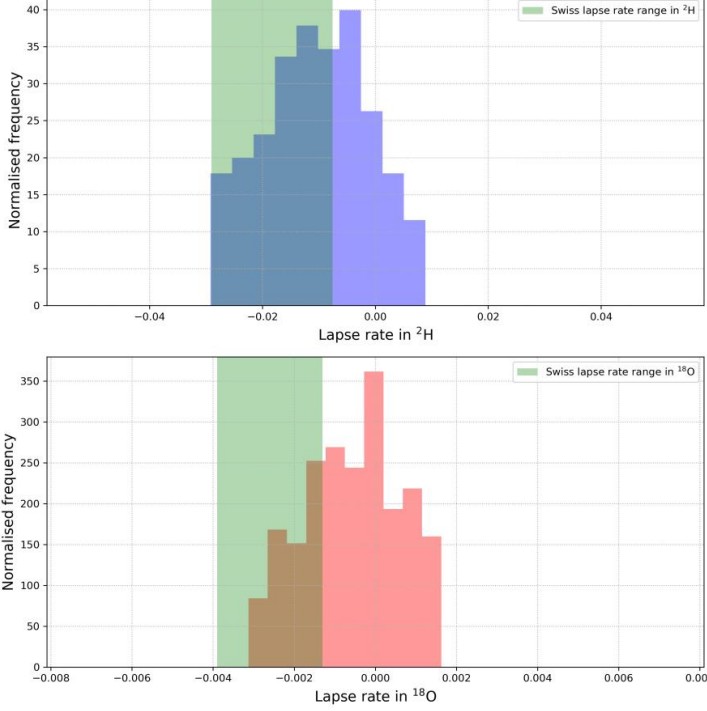

**Figure 9.** Histogram showing the posterior distribution of the isotope lapse rate parameter in
$\delta^2$H and $\delta^{18}$O. The green region shows the confidence bounds (significant at $\alpha$ =0.01) of lapse
rate computed over Switzerland by using inverse variance weighted regression. Limits of the
prior distribution of the isotopic lapse rates correspond to limits of the x-axis. The slope of the
isotopic ratio when plotted against elevation for the Swiss-wide data is shown in Figure 3 of
Beria et al. (2018).
## 5    Limitations and opportunities
As with all linear mixing models, the quality of the underlying data determines the accuracy
and utility of the results. If the tracer compositions of the different sources are not sufficiently



distinct, the uncertainty in the estimated mixing ratios will become very large. This means that if either the underlying data quality is poor, or the source contribution dynamics are not well conceptualized, then the uncertainty in the mixing ratios will be too high to be useful.

In cases where a large number of source samples are available, the computational requirements of HydroMix outweigh the benefit from using it. These are likely cases where the statistical distribution of the source tracer composition is well understood, therefore fitting a probability density curve to the source and target samples, and then inferring the distribution of the mixing ratio using a probabilistic programming approach is more appropriate (Carpenter et al., 2017; Parnell et al., 2010; Stock et al., 2018).

Finally, it is noteworthy that adding additional parameters to characterize the source tracer composition increases the degree of freedom of the model, which implies that adding such parameters leads to an increase in the uncertainty of the source contribution estimates unless new information, i.e. new observed data, is added to the model. This means that users who are interested in incorporating additional modification processes by adding parameters should ideally provide additional tracer data able to constrain this process.

For consistency and simplicity, the case studies and synthetic hydrological examples provided here focused on the contribution of rain and snow in recharging groundwater. However, it is important to emphasize that the opportunities to implement HydroMix extend to all cases where mixing contributions are of interest, and where it is difficult to build extensive databases of source tracer compositions. Such examples include quantifying the amount of "pre-event" vs. "event water" in streamflow, where "pre-event water" refers to groundwater and "event water" refers to rainfall or snowmelt. Another interesting use case might be to quantify the proportion of streamflow coming from the different source areas in a catchment, to capture the spatial dynamics of streamflow. Other uses include quantifying the amount of fog contributing to throughfall, the proportion of glacial melt vs. snowmelt flowing into a stream, the amount of vegetation water use from soil moisture at different depths vs groundwater, the interaction between surface water and groundwater at the hyporheic zone, sediment fingerprinting in fluvial systems, etc. In all of these cases, understanding source water contributions, both spatially and temporally, will improve the physical understanding of the system.

## 6   Conclusions

We develop a new Bayesian modeling framework for the application of tracers in mixing models. The primary application target of this framework is hydrology, but it is by no means limited to this field. HydroMix formulates the linear mixing problem in a Bayesian inference framework that infers the model parameters based on differences between observed and modelled tracer concentrations in the target mixture, using all possible combinations between all source and target concentration samples. For data scarce environments, this represents an advance over existing probabilistic mixing models that compute mixing ratios based on the formulation of probability distribution functions for the source and target tracer concentrations. HydroMix also makes the inclusion of additional model parameters to account



for source modification processes straightforward. Examples include known spatial or temporal tracer variations (e.g. isotopic lapse rates or evaporative enrichment).

An evaluation of HydroMix with data from different synthetic and field case studies leads to the following conclusions:

1. HydroMix gives reliable results for mixing applications with small sample sizes. As expected, the variance in source tracer composition and the ensuing composition overlap determines the uncertainty in the mixing ratio estimates. The uncertainty in mixing ratio estimates increases with increasing variance in source tracer compositions. Mixing ratio estimates improve (in terms of lower error) with increasing number of source samples.

2. As revealed by our synthetic case study with a conceptual hydrological model, at low source contributions (i.e. < 20%), a strong divergence between the actual and estimated mixing ratios emerges. This arises if HydroMix assigns equal weights to all source samples proportionally oversampling the less abundant source, which then leads to significant biases in mixing estimates. This problem is inherent to all mixing approaches, and to our knowledge has not been adequately addressed in the literature.

3. The use of composite likelihoods to weight samples by their amounts can significantly reduce the bias in the mixing estimates. At low source proportions, the estimated mixing ratio improves by more than 50% after accounting for the amount of all the sources. We show this using a simple normalization based weighting function. Future studies should explore the usage of different weighting functions that have been proposed in the past (Vasdekis et al., 2014).

4. A synthetic application of HydroMix to understand the amount of snowmelt induced groundwater recharge, revealed that using snowfall isotopic ratio instead of snowmelt isotopic ratio leads to similar mixing ratio estimates. This is particularly useful in high mountainous catchments, where sampling snowmelt is logistically difficult.

5. A real case application of HydroMix in a Swiss Alpine catchment (Vallon de Nant) showed a clear winter bias in groundwater recharge. About 60-62% of the groundwater is recharged from snowmelt (unweighted mixing approach), when snowfall only accounts for 40-45% of the total annual precipitation. This has also been previously suggested elsewhere in the European Alps (Cervi et al., 2015; Penna et al., 2014, 2017; Zappa et al., 2015).

To conclude, HydroMix provides a Bayesian approach to mixing model problems in hydrology that takes full advantage of small sample sizes. Future work will show the full potential of this approach in hydrology as well as other environmental modelling applications.

## 7 Appendix

The equations below show the numerical implementation of the evolution of isotopic ratios in snowpack and groundwater at a daily timestep.





$$C_s(t) = \frac{C_s(t-1)h_s(t-1) + C_p(t)P_s(t) - C_s(t-1)M_s(t)}{h_s(t-1) + P_s(t) - M_s(t)} \qquad 26$$

$$C_g(t) = \frac{C_g(t-1)G(t-1) + C_p(t)R_rP_r(t) + C_s(t)R_sM_s(t) - C_g(t-1)Q(t)}{G(t-1) + R_rP_r(t) + R_sM_s(t) - Q(t)} \qquad 27$$

**Author contributions**

The paper was written by HB with contributions from all co-authors. HB and BS formulated the conceptual underpinnings of HydroMix. JRL helped in framing the statistical and hydrological tests to evaluate HydroMix. AM and NCC helped in compiling data used for model evaluation and provided critical feedback during model validation.

**Code and data availability**

The model code is implemented in python 2.7 and available on GitHub at the following link https://github.com/harshberia93/HydroMix. The synthetically generated time series used in Sections 4.1, 4.2 and 4.3, along with the hypsometric curve for Vallon de Nant used in Section 4.4 are available with the model code on GitHub. The isotope data used in Section 4.4 will be made available on request.

**Acknowledgements**

The work of the authors is funded by the Swiss National Science Foundation (SNSF), grant number PP00P2\_157611. We also would like to thank Lionel Benoit for his inputs on the formulation of the Bayesian mixing model.





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
