# Peer review of "Geosci. Model Dev. Discuss., https://doi.org/10.5194/gmd-2019-69 Manuscript under review for journal Geosci. Model Dev. Discussion started: 28 March 2019"

_Geoscientific Model Development, 2019_

## Short Comment (SC1) · 15 May 2019

I am writing as an executive editor of GMD to highlight two issues with the code availability section which needs to be remedied in the revised manuscript.

Code and data only on GitHub

Thank you for providing a reference to the code and some of the data used in the experiments presented in your manuscript. There are two problems with providing this data via GitHub. The first is that a reader cannot identify the exact version of the code

that was used in the paper (for example, you may fix bugs or add features in the future). The second issue is that projects sometimes change the revision control system they use, or the hosting (the project might move to GitLab, for example). The solution to both of these issues is to provide a reference to a persistent archive of the exact version of the code that was used in the manuscript. This reference can, and should, be in addition to the GitHub link, so that a user can also always access the most recent version of the code. Since your original code is hosted on GitHub, the easiest way to produce a persistent archive of a precise version is to use GitHub's Zenodo integration. For more details, see: https://guides.github.com/activities/citable-code/.

Data only available on request

Some of the data used in the experiments is only available on request from the authors. Because authors may not always have access to this data (for example they may move institution, or data may be deleted), this is not usually an acceptable practice. This data should also be archived in a public, persistent archive such as Zenodo. It is only in the case where the authors legally cannot publish this data (for example because they only have it under licence and the licence does not permit republication) that it would be acceptable to supply data only on request. In these cases, the reason why the data cannot be published must be given in the manuscript, and the data must be identified with sufficient detail that it will always be possible for the precise data used to be identified.

---

## Author Comment (AC1) · 21 May 2019

Thank you for taking time to give your comments. There are two parts to your comment which we have addressed below (in bold original comment, below our response).

**Provide a reference to a persistent archive of the exact version of the code that was used in the manuscript**

Currently, the source code is only available on GitHub. As suggested by the executive editor, we will make the current version of the code available on Zenodo where it will be permanently archived and include the link of the same in the revised manuscript.

[Figure]

Also, we will keep the current GitHub repository active where future users can directly use the most updated version of the source code.

**Some of the data used in the experiments is only available on request from the authors. Because authors may not always have access to this data (for example they may move institution, or data may be deleted), this is not usually an acceptable practice. This data should also be archived in a public, persistent archive such as Zenodo**

As suggested, we will share the entire database used in this study through Zenodo with the source code and provide a link to the same in the revised manuscript.

---

## Referee Comment (RC1) · Anonymous Referee #1 · 26 Jun 2019

General comments

The discussed paper is well written and follows a clear and logical structure. Mixing problems are an important area of research and potentially relevant for many applications, which is why the manuscript is of potential interest to the readers of this journal. I highly appreciate the effort of the authors to contribute to this problem, and I think that Bayesian methods are well suited to address the issue. However, I have major concerns regarding some aspects of the methodological approach of this paper. Those concerns would need to be clarified and, if necessary, corrected and the results need to be re-evaluated before I can recommend publication of this manuscript. Furthermore, the manuscript lacks somewhat in clarity and rigor (see the specific comments).

Specific comments

Pg2Line30: Are there really "n" linear Equations? Since k is the number of tracers, I would assume that we have a system of k linear equations. Then, if k=n we have "n" equations and "n" unknown variables. Why "n-1"?

Pg3Line14: The authors state that it is a major shortcoming of traditional mixing models that the source concentrations are assumed to come from standard statistical distributions, which are described by some parameters. The authors should acknowledge that this can be a useful approach to account for the fact that the measurements of tracer concentrations in the same source are related to each other in some way, which is a very reasonable assumption. It is a priori not entirely clear that omitting such an assumption is beneficial to solving the mixing problem, since we might neglect reasonable prior knowledge in that case. Furthermore, the authors do not directly compare the performance of their approach to the approaches they criticize (e.g. in a synthetic case study). The added value remains therefore rather vague.

Pg3Line20-25: Arguments 1) and 2) seem to be very similar, if not the same. The authors should either provide two more distinct wordings, or combine the two arguments into one. Argument 3) is not entirely clear. Of course, the true mean and variance of the entire population can only be estimated with high uncertainty from a small sample. But this can be formally considered and should not pose a fundamental problem.

Pg3Line37: Please specify what is exactly meant by the "above limitations". The list 1)-3)? Or additional things in the text above?

Pg3Line37-46: The authors claim that the most important advantage of HydroMix is that there is no assumption about the distribution of the source tracer concentrations, if I understand correctly. While technically true, I think that this argument is misleading. Also HydroMix makes an assumption about the probabilistic nature of the model,
namely that the residuals are normally distributed with mean zero and constant variance. It is not clear to me why this should be a milder / better assumption than the one about the observations of the source concentrations being realizations of e.g. a normal distribution. All the uncertainty is just treated in a lumped way, by epsilon, which implicitly contains the deviations of the observed and the true source concentrations. Therefore, the assumptions on the source concentration distribution are not really avoided, they are just all "hidden away" in epsilon.

Pg3Line44-46: unclear what the authors mean here

Pg5Equ4: I don't understand why Eq. 4 is a solution to limitation ii).

Pg5Line11: Timestep of what? Explain what you mean by "assuming a timestep". Isn't the timestep given by the times at which the samples were taken (observed)? I don't see why "tau" can be neglected for short and long timesteps.

Pg5Equ5: what do you mean exactly by time-integrated processes? S is a state, not a process, I believe. Please clarify. You might need to provide an equation to clarify which quantity is integrated and what the lower and upper limits of integration are.

Pg5Equ6: how do "i" and "j" relate to "t"? "t" seems to disappear in the following equations. Is it reasonable to compare all the samples of the sources to all the samples of the target mixture? The authors should expand on this. When does it make sense and when not?

Pg6Line1: I believe that the citation provided here is not justified. The cited paper has nothing to do with mixing problems, it is about spectral domain likelihoods for modeling streamflow. However, it would be important to have a reference here that justifies the assumption of the Gaussian distribution for the errors in the specific case of how it is applied in this study (comparing all the measurements of source to all measurements of the mixture concentrations with normal errors). An alternative would be to check the statistical characteristics of the resulting "epsilon" and see if the normal assumption

was justified.

Pg6Line29: Please provide the original reference for importance sampling.

Pg6Line30: Please cite original reference for the Metropolis algorithm.

Pg6Line31-33: This sentence is not entirely correct and can be omitted.

My most important comment refers to Page 6 Line 42 – Page 7 Line4. The approach that the authors chose to sample from the posterior distribution is not a valid approach. Random sampling of the parameter space with retaining the best X % of the likelihood function does not yield the correct posterior distribution. Instead, the authors obtain some arbitrary measure of spread of the parameters and neither the parameter range nor the predictions done with them have any probabilistic interpretation. This is also one of the potential reasons why they do not manage to reproduce the known "rho" in Figure 2. The true value of "rho" should be inside the confidence limits also for the overlapping (high-variance) case. Also, the higher uncertainty of the mixing ratio "rho" should be visible in the high variance case, but the estimated "rhos" seem to have the same or even less uncertainty associated to them in the high variance case than in the low variance case. This seems odd to me. I would recommend that the authors implement a proper MCMC sampler (e.g. Metropolis) to obtain an actual sample from the posterior distribution. Also, the convergence of the chains needs to be checked, either by visual assessment or by convergence tests. Only converged results should be reported, otherwise meaningful conclusions are not possible.

Pg8Eq15: Does this consider rain on snow events? This might not be so important but could be mentioned.

Pg9Line26: When are samples taken in the model? How many of them are taken? Section 4.1: From the design of the case study, it is not clear if HydroMix is a statistically coherent framework. The authors should provide a proof of concept instead of or in addition to Section 4.1. The basic design of the experiment could be similar to 4.1,

but it should be done with a large number of samples, to demonstrate that HydroMix converges to the correct solution in that case. This should also be possible for the high-variance case, I think. As I mentioned before, this should be done via proper MCMC sampling, and convergence needs to be checked. For a large number of samples, the uncertainty intervals should contain the true value of "rho", also in the high variance case.

Conclusions

As it stands, conclusion 1 is favorable and it would indicate that HydroMix is a statistically valid method, but it is not supported by the results. If one replaces the word "uncertainty" by "bias", then the conclusion is supported by the results, but it is not a favorable conclusion anymore and indicates major deficiencies of the used approach. The authors should aim to obtain results that support conclusion 1.

---

## Referee Comment (RC2) · Anonymous Referee #2 · 2 Jul 2019

With the submitted manuscript, Beria et al present a new framework for end-member mixing analysis that deals with an explicit treatment of involved uncertainties using Bayesian methods. Instead of using traditional Gaussian error propagation that assumes stationary probability distributions, this Bayesian framework relies on likelihood estimation techniques that are commonly used for hydrological model parameter estimation. After deriving the theory of the approach, the authors apply their new framework on a set of case studies starting with synthetically derived data up to a real mountainous catchment in the Swiss Alps.

The study addresses a well-known problem in hydrograph separation, which is the

treatment of uncertainty, especially when only small samples sizes are available. The proposed Bayesian framework seems to provide a promising direction to provide estimates with reliable uncertainty quantification. For these reasons, I believe that this paper will make a valuable contribution to Geoscientific Model Development after some minor remarks have been addressed:

1. There is heavy referencing especially at the introduction, which sometimes appears to be inadequate (see comments in attached manuscript). Also, some literature on uncertainty in hydrograph separation is missing.

2. Some work is necessary at section 2 (Model description and implementation) to add more clarify and structure to this section. It is hard to understand what the authors are doing here.

3. Do you really need so many case studies? They make the paper long and heavy. If you don't need them to make your point, please reduce to 1 or two of them.

Some more specific and technical comments are provided in the attached pdf.

Please also note the supplement to this comment:
https://www.geosci-model-dev-discuss.net/gmd-2019-69/gmd-2019-69-RC2-supplement.pdf

———————————————————————

[Figure]

**Supplement:**

[revised manuscript text omitted]

---

## Referee Comment (RC3) · Anonymous Referee #3 · 10 Jul 2019

General comments: The study of Beria et al highlights the importance of isotope mixing modelling to overcome spatio-temporal variability inherent in end-member sampling data. To underline its use, four case studies are assigned to evaluate the modelling performance. The results underline that modelling approaches can help to better understand uncertainities and the constraints of sampling number against flux magnitudes related to runoff components. The study is an important contribution to the isotope hydrology and maybe of use also for other disciplines. Beside few specific comments, two major aspects need to be addressed before the manuscript is ready for acceptance:

- It remains unclear how the model is initiated and which data coming from field data

or other studies were used. In this context, I recommend to report more experimental data, on which you rely in a second step to drive your modelling approach.

- the study partly relies on the mixing model of precipitation and snow, not snowmelt. While snow and snowmelt have different isotopic signatures, it is conceptually more correct to use snowmelt for mixing models to estimate its contribution to groundwater. I strongly recommend to better justify why snow instead of snowmelt was used to infer the meltwater supply of groundwater if you cannot extend your calculations to snowmelt.

Specific comments: Page 2, Line 37: Please consider to address also the assumptions of mixing models and the corresponding violation. This aspect may also help to justify Bayesian mixing approaches you describe well. Page 5, Line 3: Please add here that the case studies you report later refer to mountainous (high-elevation) catchments. Page 5, Line 17: Please clarify the time-integrated processes you refer to. Page 8, Line 14-15: How is the sine function defined? How do you derive the amplitude and time lag? Page 8, Line 22: Please add some references to justify the temperature boundaries you have chosen? Other common boundaries are -1 to 3°C or-1.5 to -1.5, for example. Page 10, Line 39: Please clarify, to which small glaciers do these area proportions of 4.4 % and 10.1 % belong to? Page 11, Line 20: It is well known that the isotopic signature of snow is isotopically much more negative and thus much different from the one of snowmelt, which actually forms the 'true liquid' runoff component. Beside, which logistical constraints did prevent from sampling? Snowmelt sampling can easily be carried out by installing PCS collectors or grab sampling the dripping snowmelt. Table 1: What do you mean by top snowpack layer? Page 12, Line 34-35: Please rephrase and clarify here. The catchment average isotopic ratio does not simply depend on the elevation gradient, which may hold better for precipitation variabilities. But on the presence of the snowpack and where the snowpack is isothermal so that melting could start. Figure 3: What do you mean by "different random seeds"? Please see also the general comment regarding the modelling initiation. Page 15, Line

4-8: Please consider moving this paragraph to section 3.2 Table 4: How do you justify the values of precipitation isotopic lapse rate? Can you provide field data or further references on experimental data? Figure 6: How did you define the number of days on which rainfall, snowfall and snowmelt occurred? Please rephrase Pager 17, Line 17. Page 20, Line 3-59: Please rephrase. Figure 9: Please enlarge axis tick labels. Why did you use different x axis scales? Page 22, Line 7: What is small sampling number in your opinion? Page 22, Line 31: I do not see how sediment dynamics fit in here. Sediment dynamics may be coupled with specific runoff components or also decoupled.

Typing errors: Page 1, Line 29: Please change to "that effectively weight" Page 21, Line 17: Rephrase to "tracer data being available"

---

## Author Comment (AC2) · 17 Jul 2019

**Response to reviewer 2**

We would like to thank the anonymous reviewer for his/her valuable comments and important insights into the manuscript. The reviewer raised three broad concerns:

**There is heavy referencing especially at the introduction, which sometimes appears to be inadequate (see comments in attached manuscript). Also, some literature on uncertainty in hydrograph separation is missing.**
**P2L8-13, L15-17, L35-36: please remove some of the references. A good statement doesn't need more than 3 references for support...**

We will try and optimize the number of references in the introduction section. Also, we will add references relevant to hydrograph separation in the revised manuscript.

**Some work is necessary at section 2 (Model description and implementation) to add more clarify and structure to this section. It is hard to understand what the authors are doing here.**
**P4L10 (Section 2): Some serious work is necessary to add more clarify and structure to this section. It is hard to understand what the authors are doing here...**

We will move some of the discussion in the introduction section about linear mixing problems (P2L19-31) to this section to provide a context behind the theoretical underpinnings of HydroMix. We will also modify text within the model description section to simplify the explanation in the revised manuscript.

**Do you really need so many case studies? They make the paper long and heavy. If you don't need them to make your point, please reduce to 1 or two of them.**
**P7L13-15: Do you really need 5 case studies? They make the paper long and heavy. If you don't need them to make your point, please reduce to 1 or two of them...**

The first case study evaluates whether HydroMix converges to the correct results in standard statistical tests. As also suggested by reviewer 1, we will modify the first case study into a benchmarking test to prove the validity of the new mixing approach. The second case study evaluates HydroMix using a conceptual hydrologic model and highlights a key deficiency in commonly used mixing approaches. The third case study shows how to account for the deficiency identified earlier. The fourth case study uses HydroMix in a real case study, with the fifth one showing the flexibility offered by this HydroMix to infer additional model parameters. We feel that it is important to demonstrate both the reliability and flexibility of HydroMix. We will make this clearer in the revised manuscript.

Responses to the specific comments are mentioned below:

**P2L23-L31, P3L4-12: too methodological for an introduction**

We will replace Eq. 1 with an in-line description and move the equation to the methods section in the revised manuscript. For P3L4-12, we will try and condense the discussion with

the key insights from the studies, instead of providing a detailed explanation in the introduction.

**P3L37-38: Some improvement is necessary to provide a proper state-of-the-art with more information on what was exactly done in the many references provided above. Also, here seems to be some missing literature on methods that quantify uncertainty of mixing models. There should be many of them as first studies have been published in the 90s already:**
**Genereux, D., 1998. Quantifying uncertainty in tracer-based hydrograph separations. Water Resour. Res. 34, 915–919. doi:10.1029/98wr00010**

We agree with the reviewer that we have not done an exhaustive review of tracer-based hydrograph separation studies because hydrograph separation is only one of the common applications of mixing models used in hydrology. HydroMix is meant for applications beyond the classic hydrograph separation. However, we do understand that it might be useful to include studies spanning various hydrograph separation techniques and we will include a short overview of the same in the revised manuscript.

**P4L30: The time lag only accounts for advection right? what about dispersion or retardation?**

The time lag is meant to simulate advection, we assume that the source components are conservative in nature and do not account for dispersion as that lies beyond the scope of this paper. We will clarify this in the revised manuscript.

**P4L42: these studies use the storage selection functions approach, which is somewhat different from the approach introduced here. How do they link to this list?**

The time scale for subsurface flow strongly depends on the catchment moisture conditions. The studies cited (Benettin et al., 2017; Harman, 2015) show that the fraction of young water in streamflow depend on catchment moisture conditions, often referred to as the "inverse storage effect", i.e. there is more event water in the stream when catchment soil moisture is high. That is what we want to say. We will try and clarify this in the revised manuscript.

**P6L29-40: This is a small state-o-the-art which rather belongs to the introduction. Please keep the methods section as clear as possible.**

This section discusses the different approaches for inferring model parameters, which is why we mention this in the "*Parameter inference in a Bayesian framework*" section.

**Reference:**

Benettin, P., Bailey, S. W., Rinaldo, A., Likens, G. E., McGuire, K. J., & Botter, G. (2017). Young runoff fractions control streamwater age and solute concentration dynamics. *Hydrological Processes*, *31*(16), 2982–2986. https://doi.org/10.1002/hyp.11243
Harman, C. J. (2015). Time-variable transit time distributions and transport: Theory and

application to storage-dependent transport of chloride in a watershed. *Water Resources Research*, *51*(1), 1–30. https://doi.org/10.1002/2014WR015707

---

## Author Comment (AC3) · 17 Jul 2019

**Response to reviewer 3**

We would like to thank this anonymous reviewer for his/her valuable comments and detailed insights into the manuscript. The reviewer raises two major points stated below:

**It remains unclear how the model is initiated and which data coming from field data or other studies were used. In this context, I recommend to report more experimental data, on which you rely in a second step to drive your modelling approach.**
**Figure 3: What do you mean by "different random seeds"? Please see also the general comment regarding the modelling initiation.**

HydroMix is initiated by setting a prior distribution for the model parameters. HydroMix requires concentration data for the different components that mix linearly. In the revised version we will make clearer what kind of data is used as input for HydroMix.

In order to sample the prior distribution of the model parameters, a random number generator is required. To make the results reproducible, it is a recommended practice to specify what is sometimes called the random seed, i.e. a number used to set the random number generator to a specified state, which results in a reproducible set of random numbers (the same seed will give the same random numbers). This will be clarified in the revised version.

As also suggested by the Executive editor, we will provide the experimental data used for the Vallon de Nant case study through zenodo and include a link of the same in the revised manuscript.

**The study partly relies on the mixing model of precipitation and snow, not snowmelt. While snow and snowmelt have different isotopic signatures, it is conceptually more correct to use snowmelt for mixing models to estimate its contribution to groundwater. I strongly recommend to better justify why snow instead of snowmelt was used to infer the meltwater supply of groundwater if you cannot extend your calculations to snowmelt.**
**P11L20: It is well known that the isotopic signature of snow is isotopically much more negative and thus much different from the one of snowmelt, which actually forms the 'true liquid' runoff component. Beside, which logistical constraints did prevent from sampling? Snowmelt sampling can easily be carried out by installing PCS collectors or grab sampling the dripping snowmelt**

We agree with the reviewer's point on the usage of snowmelt isotopic ratio instead of snowfall or snowpack isotopic ratio as it is the water from snowmelt that infiltrates and recharges groundwater. However, a recent review of previous investigations that explored the differences in isotopic ratio between snowfall and snowmelt by Beria et al., (2018) revealed that snowfall and snowmelt isotopic ratios have similar mean values but different variances, with lower variance in snowmelt than snowfall isotopic ratios. Snowfall isotopic ratio was found to be the most variable followed by snowpack and then snowmelt isotopic ratio. The lower variance in snowmelt isotopic ratios is because of snowpack homogenization caused by mixing of liquid meltwater between different snowpack layers due to diffusion and dispersion.

There also exists a strong temporal variability in the meltwater isotopic ratios, with more negative isotopic ratios in the early part of the melt season (also referred to as the 'melt-out effect'), and less negative isotopic ratios in the later part of the melt season. However, on an aggregate basis, the mean values of the snowfall and snowmelt isotopic ratios are similar, except in regions with substantial sublimation, which is not the case in the Swiss Alps. This makes snowfall isotopic ratios a reasonable proxy for snowmelt isotopic ratios, which is also supported by results of the synthetic case study (Figure 6). We use snowpack isotopic ratio in the mixing case study because Vallon de Nant is a remotely located catchment with very limited winter access. The catchment experiences frequent winter avalanches and mudslides ("https://www.20min.ch/schweiz/news/story/Schlammlawine-begraebt-Strasse-unter-sich-30186734," 2019), making it hard to monitor during the winter season. This is why we were unable to install snowmelt lysimeters or PCS collectors to regularly sample meltwater. Also, as shown in the review paper of Beria et al., (2018), it is reasonable to replace snowmelt with snowpack isotopic ratios.

We will give some more details in the revised version.

**P3L37: Please consider to address also the assumptions of mixing models and the corresponding violation. This aspect may also help to justify Bayesian mixing approaches you describe well.**

Some limitations of the different mixing approaches were already discussed in the original Section 5 (Limitations and Opportunities). We will also include a short discussion in the Introduction section.

**P4L3: Please add here that the case studies you report later refer to mountainous (high-elevation) catchments**

Thanks for pointing this out. We will mention the Vallon de Nant case study here in the revised manuscript.

**P5L17: Please clarify the time-integrated processes you refer to**

This refers to a comment also made by reviewer 1 about the model formulation and the time step change. The model formulation will be clarified in the revised manuscript.

**P8L14-15: How is the sine function defined? How do you derive the amplitude and time lag? Table 4: How do you justify the values of precipitation isotopic lapse rate? Can you provide field data or further references on experimental data?**

The precipitation isotopic ratio time series was assumed to be sampled from a sinusoidal distribution, which is in-line with previous studies in Switzerland (Allen et al., 2018). The mean value and amplitude of the precipitation isotopic ratio closely corresponds with values obtained with the field data in Vallon de Nant. As already pointed out by the Executive editor, we will provide the used database of precipitation isotopic data through zenodo and include the link in the revised manuscript.

We used an offset value of (-π/2) in the sine function of the precipitation isotopic ratio (mentioned in Table 4). This corresponds to the commonly obtained seasonal variation of the precipitation isotopic ratio in Switzerland, where the ratio is lower (or more negative) during the winters and higher (or less negative) during the summers. Such a seasonal trend has also been reported previously (Allen et al., 2018; Beria et al., 2018).

**P8L22: Please add some references to justify the temperature boundaries you have chosen? Other common boundaries are -1 to 3_C or-1.5 to -1.5, for example.**

We will include the relevant references in the revised manuscript.

**P10L39: Please clarify, to which small glaciers do these area proportions of 4.4 % and 10.1 % belong to?**

The line mentioned by the reviewer reads: "*Despite the relatively low elevation, there is a small glacier with an extended moraine that covers 4.4% and 10.1% of the catchment area*"

Vallon de Nant has a small glacier on its South-western tip which covers around 4.4% of the catchment area, below which an extended moraine occupies 10.1% of the catchment area. We will clarify this in the revised manuscript.

**Table 1: What do you mean by top snowpack layer?**

Top snowpack layer refers to the top most layer of the snowpack, which we use as a proxy for recent snowfall as we do not sample snowfall.

**P12L34– 35: Please rephrase and clarify here. The catchment average isotopic ratio does not simply depend on the elevation gradient, which may hold better for precipitation variabilities. But on the presence of the snowpack and where the snowpack is isothermal so that melting could start.**

We agree with the reviewer that the spatial variability in precipitation isotopes is not only a function of elevation, but also depends on the source of moisture origin, cloud condensation temperature and snow metamorphism effects. The case study of lapse rate effect is a mere demonstration that HydroMix allows inference of additional parameters that can account for various physical processes that may modify precipitation isotopic ratio. We will clarify this in the revised manuscript.

With regards to the spatial variability in snowpacks, we use a spatially lumped hydrological model and do not explicitly simulate the spatial variability in snowpack isotopic ratios. We acknowledge that snowpacks at lower elevations melt first and they have isotopic ratios which are different from higher elevation snowpacks. We do not account for this spatial heterogeneity as this lies beyond the scope of this paper. However, we will mention this as a limitation in the revised manuscript.

**P15L4-8: Please consider moving this paragraph to section 3.2**

The reviewer refers to the following text:
*"The parameters used to generate daily precipitation, air temperature and precipitation isotopic ratios for a run time of 100 years are shown in Table 4. The static volume of groundwater that does not interact directly with the stream, GC is set to 1000 mm. Figure 4 shows the resulting variation in the isotopic ratio of groundwater over the entire 100 year period, showing the system achieves a steady state condition after ~15 years of simulation"*

Thank you for the suggestion. We will move this section to section 3.2 in the revised manuscript.

**Figure 6: How did you define the number of days on which rainfall, snowfall and snowmelt occurred?**

In order to simulate precipitation, time between two successive precipitation events is modelled as a Poisson process, with the number of yearly precipitation events specified as 30. The snow accumulation and the degree-day snowmelt model are then used to compute the number of snowfall days and of snowmelt events. This will be made clear in the revised version.

**P17L17, P20L3-59: Please rephrase**

P17L17 reads:
*"In this study, we have used a relatively simple normalization based weighting function (Eq. (24)). Testing other weighting functions which have been proposed in the past (Vasdekis et al., 2014) is certainly possible, and is left for future research."*

The last sentence part is indeed not well formulated. We will rephrase it in the revised manuscript

**Figure 9: Please enlarge axis tick labels. Why did you use different x axis scales?**

We will use larger tick labels in the revised manuscript.

The x axis of the top subplot of Figure 9 shows the lapse rate in $^2$H whereas the bottom subplot shows the lapse rate in $^{18}$O. As the isotopic ratios in $^2$H and $^{18}$O are different, the lapse rates are also different, which is why the two subplots have different x-axis scales.

**P22L7: What is small sampling number in your opinion?**

Small sampling sizes can be anywhere less than 20-30 samples. We will mention this in the revised version.

**P21L31: I do not see how sediment dynamics fit in here. Sediment dynamics may be coupled with specific runoff components or also decoupled.**

Mixing models are frequently used in sediment fingerprinting to quantify the sediment contribution from different parts of a catchment. Blake et al., (2018) is a recent example

where a Bayesian mixing model was used to understand the spatial origin of river sediments. This will be made clearer in the revised version.

**Typing errors: P1L29: Please change to "that effectively weight"**
**P21L17: Rephrase to "tracer data being available"**

Thanks, will be changed.

**Reference**:

Allen, S. T., Kirchner, J. W., & Goldsmith, G. R. (2018). Predicting spatial patterns in precipitation isotope (δ2H and δ18O) seasonality using sinusoidal isoscapes. *Geophysical Research Letters*, *45*(10), 4859–4868. https://doi.org/10.1029/2018GL077458

Beria, H., Larsen, J. R., Ceperley, N. C., Michelon, A., Vennemann, T., & Schaefli, B. (2018). Understanding snow hydrological processes through the lens of stable water isotopes. *Wiley Interdisciplinary Reviews: Water*, *5*(6), e1311. https://doi.org/10.1002/wat2.1311

Blake, W. H., Boeckx, P., Stock, B. C., Smith, H. G., Bodé, S., Upadhayay, H. R., et al. (2018). A deconvolutional Bayesian mixing model approach for river basin sediment source apportionment. *Scientific Reports*, *8*(1), 13073. https://doi.org/10.1038/s41598-018-30905-9

https://www.20min.ch/schweiz/news/story/Schlammlawine-begraebt-Strasse-unter-sich-30186734. (2019). Retrieved from https://www.20min.ch/schweiz/news/story/Schlammlawine-begraebt-Strasse-unter-sich-30186734

---

## Author Comment (AC4) · 19 Jul 2019

**Response to reviewer 1**

We would like to thank this anonymous reviewer for his/her detailed reading of our manuscript and the very useful comments. One main concern raised by the reviewer is about the statistical validity of this approach, specifically the usage of importance sampling instead of an Markov chain Monte Carlo (MCMC) approach to sample the posterior distribution of the mixing ratio. As we discuss hereafter, we will implement an MCMC sampler to have a complete comparison with the method proposed. Responses to the specific comments are below.

**P2L30: Are there really "n" linear Equations? Since k is the number of tracers, I would assume that we have a system of k linear equations. Then, if k=n we have "n" equations and "n" unknown variables. Why "n-1"?**

Assuming we have "*k*" tracers, there will be "*k*+1" linear equations. The first "k" equations are:

$$\rho_1 S_1^k + \rho_2 S_2^k + \cdots + \rho_n S_n^k = Y^k. \qquad\qquad 1$$

The last equation corresponds to the aggregation of different sources in the mixture, i.e. $\sum_{i=1}^n \rho_i = 1$ , which we mention in P1L30. In order to solve this system of linear equations, "n-1" different tracers are required. As also proposed by reviewer 2, we will state this clearly in the revised manuscript and move the equation part to the methods section.

**P3L14: The authors state that it is a major shortcoming of traditional mixing models that the source concentrations are assumed to come from standard statistical distributions, which are described by some parameters. The authors should acknowledge that this can be a useful approach to account for the fact that the measurements of tracer concentrations in the same source are related to each other in some way, which is a very reasonable assumption. It is a priori not entirely clear that omitting such an assumption is beneficial to solving the mixing problem, since we might neglect reasonable prior knowledge in that case.**

We thank the reviewer for this very important comment. The reviewer correctly points out that the traditional approach of fitting a statistical distribution to the source concentrations reflects a priori knowledge, which may be useful in a lot of cases. We will mention this in the 'Limitations and Opportunities' Section, where we think this discussion is best located. The main contribution of this paper is in instances where a priori knowledge about the source concentration is limited. These are mostly cases where very few measurements of source contributions are available, with limited a priori knowledge where the usage of HydroMix is more appropriate.

**Furthermore, the authors do not directly compare the performance of their approach to the approaches they criticize (e.g. in a synthetic case study). The added value remains therefore rather vague.**

We first would like to point out that we did not mean to criticize existing approaches in our manuscript. We will carefully revise the text to downgrade such critical statements since our approach is supposed to address mixing problems that are not straightforward to address with existing methods. We indeed do not compare our approach to existing approaches in detail here, since our objective is to present a new method for data sparse situations in which existing methods are not appropriate. Instead, we simulate synthetic case studies to evaluate the results of HydroMix. As these are statistical tests, the correct answer is known a priori, and we evaluate if HydroMix converges to the correct results. Given that the paper is already rather long, as also pointed out by reviewer 2, we decided not to increase the complexity and length of the paper.

**P3L20-25: Arguments 1) and 2) seem to be very similar, if not the same. The authors should either provide two more distinct wordings, or combine the two arguments into one. Argument 3) is not entirely clear. Of course, the true mean and variance of the entire population can only be estimated with high uncertainty from a small sample. But this can be formally considered and should not pose a fundamental problem.**

The reviewer refers here to the following statements:

*1) The mean and variance may not accurately reflect the statistical properties of the source composition. 2) If there is a large amount of information on the source composition, the mean and variance may be an unnecessary simplification of its variability. 3) If the source compositions have a low number of samples, then the mean and variance estimates may be poorly constrained.*

We agree that the stated arguments look very similar. This is because we argue mostly based on the very commonly assumed normality condition for the pdf of the different source concentrations, obtained empirically with small sample sizes. We will combine the three points into one sentence in the revised manuscript.

**P3L37: Please specify what is exactly meant by the "above limitations". The list 1)-3)? Or additional things in the text above?**

We were referring to the previous two paragraphs, we will clarify this in the revised manuscript.

**P3L37-46: The authors claim that the most important advantage of HydroMix is that there is no assumption about the distribution of the source tracer concentrations, if I understand correctly. While technically true, I think that this argument is misleading. Also HydroMix makes an assumption about the probabilistic nature of the model, namely that the residuals are normally distributed with mean zero and constant variance. It is not clear to me why this should be a milder / better assumption than the one about the observations of the source concentrations being realizations of e.g. a normal distribution. All the uncertainty is just treated in a lumped way, by epsilon, which implicitly contains the deviations of the observed and the true source concentrations. Therefore, the assumptions on the source concentration distribution are not really avoided, they are just all "hidden away" in epsilon.**

This is an important comment, which we will carefully address in the revised version to avoid "hiding away" the statistical assumptions. Thanks for pointing this out. As well summarized by the reviewer, the main difference of HydroMix is that it parameterizes the error function instead of the statistical distribution of the source concentrations. Parameterizing source concentrations requires large sample sizes, which is seldom the case in tracer hydrology. Error parameterization offers a useful alternative and is interesting since our methodological set-up allows increasing the error sample size compared to the input observation sample size. Furthermore, since the model error results from the aggregation of a large number of simplifications and observational errors, i.e. it aggregates a large number of random variables, the error can reasonably be assumed (central limit theorem) to follow a normal distribution. Despite this point, and as in any comparable inference approach (based on model residuals), assessing a posteriori the validity of the underlying distributional assumptions is a key step (we will include such an assessment in the revised version).

In hydrological modeling, this model residual assessment step might typically reveal that the residuals do not have zero mean (i.e. there is a bias). It is noteworthy that the normality assumption (and namely the zero mean assumption) for the model error is not required and could be modified. And, as a side note, Bayesian parameter inference assuming a wrong error distribution has been shown to be surprisingly robust for hydrological modelling (Schaefli & Kavetski, 2017).

In addition, as the reviewer correctly states above, the proposed approach lumps all error sources into a single error term (i.e. input observational uncertainties, model uncertainties, output observational uncertainties, parameter uncertainties). This is standard in hydrological model inference (e.g. Schaefli et al., (2007)) but there is indeed a large body of literature that discusses how this could be relaxed (e.g. via explicit accounting for observational uncertainties (Kavetski et al., 2006)), with the inherent limit that we do not know how to account for model structural uncertainties (e.g. a missing source in our case). We will mention this limitation in the revised version.

**P3L44-46: unclear what the authors mean here**

The reviewer refers to the following sentence:
"*An advantage of this approach is that additional model parameters can be incorporated in the framework to describe how the source tracer concentrations might be modified according to specific hydrologic processes that can be decided and explored by the user.*"

In HydroMix, the model parameter (mixing ratio) is inferred in an inverse modeling setup. Theoretically, multiple parameters can be inferred in this framework. We demonstrate this in case study 3.5, where isotopic lapse rate is inferred along with the mixing ratio. We will make this clearer in the revised manuscript.

**P5Eq4: I don't understand why Eq. 4 is a solution to limitation ii)**

This was a typo, correct would have been "to limitation iv)".

Besides, we also remove the statement "*Section 3.5 introduces such an example and proposes a solution in these cases*." Since this cuts the flow of text.

**P5L11: Timestep of what? Explain what you mean by "assuming a timestep". Isn't the timestep given by the times at which the samples were taken (observed)? I don't see why "tau" can be neglected for short and long timesteps.**

The formulation was not clear in the paper. $t$ is indeed the sampling time. We assume however, that all observed samples are samples from the long term (>> 1 year) source and target compositions, i.e. the modelling time step is much larger than the lag, which can thus be removed from the equation. The difference between sampling time step and modelling time step will be made clear in the revised version. We will also remove the sentence that $tau$ could also be short. This was misleading.

**P5Eq5: What do you mean exactly by time-integrated processes? S is a state, not a process, I believe. Please clarify. You might need to provide an equation to clarify which quantity is integrated and what the lower and upper limits of integration are.**

We use here the term process in a generic sense to designate a stochastic process. The explanation in the text was not clear, we simply use different symbols for stochastic processes with different temporal support. This will be clarified.

**P5Eq6: how do "i" and "j" relate to "t"? "t" seems to disappear in the following equations.**

We did indeed not explicitly mention that we omit the time stamp of each sample and simply number them (thus the shift from $t$ to $i$). This happens because we do not consider our samples as being a time series but simply as being a set of samples from a random variable, the source composition. This shift from a temporal process to a time-averaged random variable was indeed not clear in the manuscript. Given the questions of this and the 2$^{nd}$ reviewer, we consider to re-formulate this entire section. It is probably not necessary to present the temporal process viewpoint before switching to the source composition viewpoint.

**Is it reasonable to compare all the samples of the sources to all the samples of the target mixture? The authors should expand on this. When does it make sense and when not?**

HydroMix does not take into account the time at which a source was sampled exactly because we assume that each sample provides a sample of a random variable, which is the source composition. To build the model residuals, we need to compare the model output (modeled target compositions) with observed samples. Assuming that all available source samples are independent, building all possible model simulations from the available source samples is the most natural solution. It is noteworthy that even if the source and target samples are taken at the same time, this does not mean a priori that they are dependent since the target samples result from water that has travelled a certain time in the catchment, i.e. target sample from a time step $t$ do not result from the source composition at the same time $t$. If in a system we have instantaneous mixing, then the source and target samples taken at the same moment in time will necessarily be strongly correlated. In this case the assumption of

independent samples and thus the combination of all samples would not make sense. We will specify all this in the revised version and try to come up with an example where the assumption of independent samples would not make sense.

**P6L1: I believe that the citation provided here is not justified. The cited paper has nothing to do with mixing problems, it is about spectral domain likelihoods for modeling streamflow. However, it would be important to have a reference here that justifies the assumption of the Gaussian distribution for the errors in the specific case of how it is applied in this study (comparing all the measurements of source to all measurements of the mixture concentrations with normal errors). An alternative would be to check the statistical characteristics of the resulting "epsilon" and see if the normal assumption was justified.**

The cited paper is indeed about spectral calibration, which is insensitive to the distribution of errors in the time-domain. Therefore it compares spectral-domain to time-domain calibration and an important result is that time-domain calibration is very robust under wrong distributional assumptions, and in particular much more robust than commonly assumed in hydrological modelling. We will discuss why Gaussianity is justified and we will assess how well the normality assumption holds in our examples.

**P6L29-30: Please provide the original reference for importance sampling and Metropolis algorithm.**

We will include them in the revised manuscript.

**P6L31-33: This sentence is not entirely correct and can be omitted**

We will remove this in the revised manuscript.

**My most important comment refers to P6L42 – P7L4. The approach that the authors chose to sample from the posterior distribution is not a valid approach. Random sampling of the parameter space with retaining the best X % of the likelihood function does not yield the correct posterior distribution. Instead, the authors obtain some arbitrary measure of spread of the parameters and neither the parameter range nor the predictions done with them have any probabilistic interpretation. This is also one of the potential reasons why they do not manage to reproduce the known "rho" in Figure 2. The true value of "rho" should be inside the confidence limits also for the overlapping (high-variance) case. Also, the higher uncertainty of the mixing ratio "rho" should be visible in the high variance case, but the estimated "rhos" seem to have the same or even less uncertainty associated to them in the high variance case than in the low variance case. This seems odd to me. I would recommend that the authors implement a proper MCMC sampler (e.g. Metropolis) to obtain an actual sample from the posterior distribution. Also, the convergence of the chains needs to be checked, either by visual assessment or by convergence tests. Only converged results should be reported, otherwise meaningful conclusions are not possible.**

We agree with the comments of the reviewer and will implement a MCMC sampler to sample the posterior function in the revised manuscript.

**P8Eq15: Does this consider rain on snow events? This might not be so important but could be mentioned**

In the hydrologic model used, heat exchange within the snowpack from rain-on-snow events are not considered as that lies beyond the scope of this paper. We will mention this in the revised manuscript.

**P9L26: When are samples taken in the model? How many of them are taken?**

In this case study, isotopic ratios are simulated and all the rain and snowmelt isotopic ratios are used with HydroMix. The number of rainfall, snowfall and snowmelt samples are 39, 24 and 107, as mentioned in the labels of Figures 6 and 7. We will also mention this in this section in the revised manuscript.

**Section 4.1: From the design of the case study, it is not clear if HydroMix is a statistically coherent framework. The authors should provide a proof of concept instead of or in addition to Section 4.1. The basic design of the experiment could be similar to 4.1, but it should be done with a large number of samples, to demonstrate that HydroMix converges to the correct solution in that case. This should also be possible for the high variance case, I think. As I mentioned before, this should be done via proper MCMC sampling, and convergence needs to be checked. For a large number of samples, the uncertainty intervals should contain the true value of "rho", also in the high variance case.**

Thank you for this very valuable comment. The current section 4.1 will be modified into a proof of concept test. In the current version of the manuscript, we have shown that the error in estimation of the mixing ratio reduces with increasing sample size (Figure 3a). In addition, we will add a test that evaluates the convergence of HydroMix for the high variance case with a large number of samples. This we believe will test the statistical validity of the approach.

**As it stands, conclusion 1 is favorable and it would indicate that HydroMix is a statistically valid method, but it is not supported by the results. If one replaces the word "uncertainty" by "bias", then the conclusion is supported by the results, but it is not a favorable conclusion anymore and indicates major deficiencies of the used approach. The authors should aim to obtain results that support conclusion 1.**

We do not fully understand the point that the reviewer is trying to make here. Conclusion 1 reads as:
*"HydroMix gives reliable results for mixing applications with small sample sizes. As expected, the variance in source tracer composition and the ensuing composition overlap determines the uncertainty in the mixing ratio estimates. The uncertainty in mixing ratio estimates increases with increasing variance in source tracer compositions. Mixing ratio estimates improve (in terms of lower error) with increasing number of source samples."*

We believe that the proof of concept test will prove the reliability and validity of HydroMix. We will revise the text according to the new results with the MCMC sampler.

---

## Author Response (AR1)

**Submission of revised version of manuscript titled "HydroMix v1.0: a new Bayesian mixing framework for attributing uncertain hydrological sources "**

We would like to thank Bethanna Jackson, David Ham, along with the three reviewers for their valuable feedback. Their inputs led to significant improvements in the manuscript.

We have incorporated the suggestions made by the reviewers in the revised manuscript. We have now implemented a Markov Chain Monte Carlo (MCMC) sampler to infer the posterior distribution of the model parameters. As recommended by Executive editor David Ham, we have made the model along with the datasets used through zenodo, which is available at http://doi.org/10.5281/zenodo.3475429.

We hope that with these additions and all modifications discussed below, the revised manuscript is now ready for publication in Geoscientific Model Development.

Regards,
Harsh Beria

**Response to reviewer 1**

We would like to thank this anonymous reviewer for his/her detailed reading of our manuscript and the very useful comments. One main concern raised by the reviewer is about the statistical validity of this approach, specifically the usage of importance sampling instead of an Markov chain Monte Carlo (MCMC) approach to sample the posterior distribution of the mixing ratio. We implemented an MCMC sampler as discussed hereafter. Responses to the specific comments are below. The reviewer comments are in bold, answers in normal style and the quoted text from the revised manuscript in italics.

**P2L30: Are there really "n" linear Equations? Since k is the number of tracers, I would assume that we have a system of k linear equations. Then, if k=n we have "n" equations and "n" unknown variables. Why "n-1"?**

Assuming we have "*k*" tracers, there will be "*k*+1" linear equations. The first "k" equations are:

$$\rho_1 S_1^k + \rho_2 S_2^k + \cdots + \rho_n S_n^k = Y^k. \qquad\qquad 1$$

The last equation corresponds to the aggregation of different sources in the mixture, i.e. $\sum_{i=1}^{n} \rho_i = 1$. In order to solve this system of linear equations, "n-1" different tracers are required.

In order to provide more clarity as also proposed by Reviewer 2, we moved Eq. 1 to the beginning of Section 2 (Model description and implementation) (P3L43-P4L4) in the revised manuscript. We also clarified the text which now reads as:

*"A system with n sources mixing linearly in a target water body can be written as:*

$$\rho_1 S_1^k + \rho_2 S_2^k + \cdots + \rho_n S_n^k = Y^k, \qquad\qquad 2$$

*where $Y^k$ is the concentration of the $k^{th}$ tracer in the target mixture, $S_i^k$ is the concentration of the $k^{th}$ tracer in source i. $\rho_i$ (i=1, .., n) are the fractions of all sources in the mixture, with $\sum_{i=1}^{n} \rho_i = 1$, corresponding to the aggregation of different sources in the mixture. In order to solve this system of linear equations, "n-1" different tracers are required."*

**P3L14: The authors state that it is a major shortcoming of traditional mixing models that the source concentrations are assumed to come from standard statistical distributions, which are described by some parameters. The authors should acknowledge that this can be a useful approach to account for the fact that the measurements of tracer concentrations in the same source are related to each other in some way, which is a very reasonable assumption. It is a priori not entirely clear that omitting such an assumption is beneficial to solving the mixing problem, since we might neglect reasonable prior knowledge in that case.**

We thank the reviewer for this very important comment. The reviewer correctly points out that the traditional approach of fitting a statistical distribution to the source concentrations reflects *a priori* knowledge, which may be useful in a lot of cases. We now mention this in the 'Limitations and Opportunities' Section of the revised manuscript (P21L10-12), where we think this discussion is best located. The text reads as:

*"Also HydroMix might not be an appropriate method in instances where fitting statistical distributions to source and target compositions reflect a priori knowledge of the system."*

**Furthermore, the authors do not directly compare the performance of their approach to the approaches they criticize (e.g. in a synthetic case study). The added value remains therefore rather vague.**

We first would like to point out that we did not mean to criticize existing approaches in our manuscript. We indeed do not compare our approach to existing approaches in detail here, since our objective is to present a new method for data sparse situations in which existing methods are not appropriate. To accomplish that, we simulated synthetic case studies to evaluate the results of HydroMix (Section 4.1 of the revised manuscript). As these are statistical tests, the correct answer is known *a priori*, and HydroMix converges to the correct results as is shown in Figures 2 and 3. Given that the paper is already rather long, as also pointed out by reviewer 2, we decided not to increase the complexity and length of the paper by adding a new case study to compare the results of HydroMix with existing methods.

**P3L20-25: Arguments 1) and 2) seem to be very similar, if not the same. The authors should either provide two more distinct wordings, or combine the two arguments into one. Argument 3) is not entirely clear. Of course, the true mean and variance of the entire population can only be estimated with high uncertainty from a small sample. But this can be formally considered and should not pose a fundamental problem.**

The reviewer refers here to the following statements in the original manuscript:

*"1) The mean and variance may not accurately reflect the statistical properties of the source composition. 2) If there is a large amount of information on the source composition, the mean and variance may be an unnecessary simplification of its variability. 3) If the source compositions have a low number of samples, then the mean and variance estimates may be poorly constrained."*

We agree that the stated arguments look very similar. This is because we argue mostly based on the very commonly assumed normality condition for the pdf of the different source concentrations, obtained empirically with small sample sizes. We now combine the three points into one sentence (P3L4-L9) in the revised manuscript which reads as:

*"This limits both the potential applicability and the insights that can be gained from tracer information in hydrology because the sample mean and variance may not accurately reflect the statistical properties of the actual source composition and the Gaussian approach represents an unnecessary simplification in cases where a large amount of information on source composition is available."*

**P3L37: Please specify what is exactly meant by the "above limitations". The list 1)-3)? Or additional things in the text above?**

We were referring to the previous two paragraphs, we have clarified this in the revised manuscript (P3L20) which reads as:

*"To overcome the limitations of source heterogeneity and the previously discussed restriction to Gaussian distributions, we present a new mixing approach for hydrological applications, called HydroMix."*

**P3L37-46: The authors claim that the most important advantage of HydroMix is that there is no assumption about the distribution of the source tracer concentrations, if I understand correctly. While technically true, I think that this argument is misleading. Also HydroMix makes an assumption about the probabilistic nature of the model, namely that the residuals are normally distributed with mean zero and constant variance. It is not clear to me why this should be a milder / better assumption than the one about the observations of the source concentrations being realizations of e.g. a normal distribution. All the uncertainty is just treated in a lumped way, by epsilon, which implicitly contains the deviations of the observed and the true source concentrations. Therefore, the assumptions on the source concentration distribution are not really avoided, they are just all "hidden away" in epsilon.**

This is an important comment, which we have carefully addressed in the revised manuscript and also discussed the same in the "Limitations and opportunities" section of the revised manuscript (P21L14-21), which reads as:

*"A key difference between HydroMix and other Bayesian mixing approaches is that HydroMix parameterizes the error function whereas other Bayesian approaches parameterize the statistical distribution of source and mixture compositions. Parameterizing source compositions require large sample sizes, which is seldom the case in tracer hydrology. Error parameterization offers a useful alternative and can be also verified against the posterior error distribution. In the case studies demonstrated in this paper, a normal error model was found to be appropriate. However, error models other than Gaussian can be used by formulating the respective likelihood function."*

Parameterizing source concentrations requires large sample sizes, which is seldom the case in tracer hydrology. Error parameterization offers a useful alternative and is interesting since our methodological set-up allows increasing the error sample size compared to the input observation sample size. Furthermore, since the model error results from the aggregation of a large number of simplifications and observational errors, i.e. it aggregates a large number of random variables, the error can reasonably be assumed (central limit theorem) to follow a normal distribution. Despite this point, and as in any comparable inference approach (based on model residuals), assessing a posteriori the validity of the underlying distributional assumptions is a key step and we have included an error plot showing the evolution of mean and variance of the error in the revised version (Figure 3 in the revised manuscript, also reproduced below). Also, we have shown Q-Q plots below verifying that posterior distribution of model error closely resemble a normal distribution for different mixing ratios. Given that the paper is already rather long, as also pointed out by reviewer 2, we have not included this figure in the revised manuscript.

[Figure]

**Supplementary Figure**. Q-Q plots showing the posterior distribution of model errors for five distinct mixing ratios for the low variance dataset.

[Figure]

*Figure 1. Diagnostic plots showing the convergence characteristics of MCMC chains for five different mixing ratios for the low variance dataset (shown in Error! Reference source not found.). Subplots (a) and (b) show variations in the inferred mixing ratio and the error mean with increasing MCMC runs.*

In addition, as the reviewer correctly states above, the proposed approach lumps all error sources into a single error term (i.e. input observational uncertainties, model uncertainties, output observational uncertainties, parameter uncertainties). This is standard in hydrological model inference (e.g. Schaefli et al., (2007)) but there is indeed a large body of literature that discusses how this could be relaxed (e.g. via explicit accounting for observational uncertainties (Kavetski et al., 2006)), with the inherent limit that we do not know how to account for model structural uncertainties (e.g. a missing source in our case).

**P3L44-46: unclear what the authors mean here**

The reviewer refers to the following sentence:
"*An advantage of this approach is that additional model parameters can be incorporated in the framework to describe how the source tracer concentrations might be modified according to specific hydrologic processes that can be decided and explored by the user.*"

In HydroMix, the model parameter (mixing ratio) is inferred in an inverse modeling setup. Theoretically, multiple parameters can be inferred in this framework. We demonstrate this in case study 3.5, where isotopic lapse rate is inferred along with the mixing ratio. We have clarified this in the revised manuscript (P3L27-29) which reads as:

"*Multiple model parameters can be inferred in such a setup allowing parameterization of additional hydrologic processes that can modify source tracer concentrations (shown in Section **Error! Reference source not found.**).*"

**P5Eq4: I don't understand why Eq. 4 is a solution to limitation ii)**

This was a typo, correct would have been "to limitation iv)" (P5L1). We have rectified it in the revised manuscript. Besides, we also removed the statement "*Section 3.5 introduces such an example and proposes a solution in these cases*" since it cuts the flow of text.

**P5L11: Timestep of what? Explain what you mean by "assuming a timestep". Isn't the timestep given by the times at which the samples were taken (observed)? I don't see why "tau" can be neglected for short and long timesteps.**

The formulation was not clear in the paper. $t$ is indeed the sampling time. We assume however, that all observed samples are samples from the long term (>> 1 year) source and target compositions, i.e. the modelling time step is much larger than the lag, which can thus be removed from the equation. We clarified this in the revised manuscript (P5L7-L10). We also removed the sentence that *tau* could also be short as it was misleading. The new text read as:

"*Limitation iii) can be relaxed by assuming a long enough timestep (eg: long term groundwater recharge dynamics), where the observed samples are samples from the long term (>> 1 year) source and target compositions. This allows to replace the timestep 't' and 't+ τ' with Δt and write Eq. (2) as:*"

**P5Eq5: What do you mean exactly by time-integrated processes? S is a state, not a process, I believe. Please clarify. You might need to provide an equation to clarify which quantity is integrated and what the lower and upper limits of integration are.**

We use here the term process in a generic sense to designate a stochastic process. To clarify the explanation, we dropped the word "process" (P5L14-L19).The new text reads as:.

*"where the ' signifies the new time-integrated variables. Now, any observed point-scale tracer concentration $p_i$ in a given source i or in the output (e.g., the isotopic ratio of snowmelt) can be assumed to represent a sample from a stationary process (from $S'_1$ or $S'_2$ or $Y'$),. This assumption is in fact implicitly underlying most of the existing hydrological mixing models where point samples are used to characterize a spatial process and where the time reference of the samples is discarded."*

**P5Eq6: how do "i" and "j" relate to "t"? "t" seems to disappear in the following equations.**

We did indeed not explicitly mention that we omit the time stamp of each sample and simply number them (thus the shift from *t* to *i*). This happens because we do not consider our samples as being a time series but simply as being a set of samples from a random variable, the source composition. As also stated in the reply to the previous comment, this has now been clarified in the revised manuscript (P5L14-L19).

*"where the ' signifies the new time-integrated variables. Now, any observed point-scale tracer concentration $p_i$ in a given source i or in the output (e.g., the isotopic ratio of snowmelt) can be assumed to represent a sample from a stationary process (from $S'_1$ or $S'_2$ or $Y'$),. This assumption is in fact implicitly underlying most of the existing hydrological mixing models where point samples are used to characterize a spatial process and where the time reference of the samples is discarded."*

**Is it reasonable to compare all the samples of the sources to all the samples of the target mixture? The authors should expand on this. When does it make sense and when not?**

HydroMix does not take into account the time at which a source was sampled exactly because we assume that each sample provides a sample of a random variable, which is the source composition. To build the model residuals, we need to compare the model output (modeled target compositions) with observed samples. Assuming that all available source samples are independent, building all possible model simulations from the available source samples is the most natural solution. It is noteworthy that even if the source and target samples are taken at the same time, this does not mean *a priori* that they are dependent since the target samples result from water that has travelled a certain time in the catchment, i.e. target sample from a time step *t* do not result from the source composition at the same time *t*. If in a system we have instantaneous mixing, then the source and target samples taken at the same moment in time will necessarily be strongly correlated. In this case the assumption of independent samples and thus the combination of all samples would not make sense. We have specified this in the "Limitations and opportunities" section of the revised version (P21L23-L31).

*"HydroMix builds the model residuals by comparing all the observed source samples with all the observed samples of the target mixture, assuming that all available source and target samples are independent. Interestingly, the assumption of independence holds even if the source and target samples are taken at the same time, since the target samples result from water that has travelled for a certain amount of time in the catchment, and hence is not related to the water entering the catchment. However, if a system has instantaneous mixing, then the source and target samples taken at the same moment of time will necessarily be strongly correlated. In such cases, the assumption of independent samples would not make sense and the method might give spurious results."*

**P6L1: I believe that the citation provided here is not justified. The cited paper has nothing to do with mixing problems, it is about spectral domain likelihoods for modeling streamflow. However, it would be important to have a reference here that justifies the assumption of the Gaussian distribution for the errors in the specific case of how it is applied in this study (comparing all the measurements of source to all measurements of the mixture concentrations with normal errors). An alternative would be to check the statistical characteristics of the resulting "epsilon" and see if the normal assumption was justified.**

The cited paper is indeed about spectral calibration, which is insensitive to the distribution of errors in the time-domain. Therefore it compares spectral-domain to time-domain calibration and an important result is that time-domain calibration is very robust under wrong distributional assumptions, and in particular much more robust than commonly assumed in hydrological modelling. We also show the Q-Q plots below verifying that posterior distribution of model error closely resemble a normal distribution for different mixing ratios.

[Figure]

**Supplementary Figure**. Q-Q plots showing the posterior distribution of model errors for five distinct mixing ratios for the low variance dataset.

**P6L29-30: Please provide the original reference for importance sampling and Metropolis algorithm.**

We have include the original references in the revised manuscript (P6L26-L27).

**P6L31-33: This sentence is not entirely correct and can be omitted**

We have removed this sentence in the revised manuscript.

**My most important comment refers to P6L42 – P7L4. The approach that the authors chose to sample from the posterior distribution is not a valid approach. Random sampling of the parameter space with retaining the best X % of the likelihood function does not yield the correct posterior distribution. Instead, the authors obtain some arbitrary measure of spread of the parameters and neither the parameter range nor the predictions done with them have any probabilistic interpretation. This is also one of the potential reasons why they do not manage to reproduce the known "rho" in Figure 2. The true value of "rho" should be inside the confidence limits also for the overlapping (high-variance) case. Also, the higher uncertainty of the mixing ratio "rho" should be visible in the high variance case, but the estimated "rhos" seem to have the same or even less uncertainty associated to them in the high variance case than in the low variance case. This seems odd to me. I would recommend that the authors implement a proper MCMC sampler (e.g. Metropolis) to obtain an actual sample from the posterior distribution. Also, the convergence of the chains needs to be checked, either by visual assessment or by convergence tests. Only converged results should be reported, otherwise meaningful conclusions are not possible.**

We agree with the comments of the reviewer and have implemented a MCMC sampler to sample the posterior function in the revised manuscript.

**P8Eq15: Does this consider rain on snow events? This might not be so important but could be mentioned**

In the hydrologic model used, heat exchange within the snowpack from rain-on-snow events are not considered as that lies beyond the scope of this paper. We have mention this in the revised manuscript (P8L31-L32):

*"Rain-on-snow events are not explicitly considered as this lies beyond the scope of this paper."*

**P9L26: When are samples taken in the model? How many of them are taken?**

In this case study, isotopic ratios are simulated and all the rain and snowmelt isotopic ratios are used with HydroMix. The number of rainfall, snowfall and snowmelt samples are 39, 24 and 107, as mentioned in the labels of Figures 6 and 7.

**Section 4.1: From the design of the case study, it is not clear if HydroMix is a statistically coherent framework. The authors should provide a proof of concept instead of or in addition to Section 4.1. The basic design of the experiment could be similar to 4.1, but it should be done with a large number of samples, to demonstrate that HydroMix converges to the correct solution in that case. This should also be possible for the high variance case, I think. As I mentioned before, this should be done via proper MCMC sampling, and convergence needs to be checked. For a large number of samples, the uncertainty intervals should contain the true value of "rho", also in the high variance case.**

Thank you for this very valuable comment. We have now implemented a MCMC sampler and included a model diagnostic plot showing convergence of the model parameters along with the variation in error mean (Figure 3 in the revised manuscript). We have reproduced figure 2, which reinstates that error decreases with increasing sample size. We added a case with a sample size of 100, where there was a sharp drop recorded in the error statistic when compared to the original simulation size of 40. We did not perform simulations for sample sizes larger than 100 as the computational requirement increases exponentially with increasing sample sizes. Additionally, figure 2b clearly shows decreasing error with increasing sample size and figure 3 shows that the MCMC chains converge rather quickly. This we believe tests the statistical validity of the approach. The figures have been reproduced below:

[Figure]

**Figure 2.** *(a) Scatterplot showing the mixing ratio (ρ) values inferred using HydroMix for the low and high variance synthetic case of* **Error! Reference source not found.***. The number of source and target samples are 100. (b) Performance of HydroMix in terms of the absolute error between the posterior mixing ratio mean and the true mean for the low variance dataset, summed over all tested ratios plotted as a function of the number of samples drawn for the two sources.*

[Figure]

***Figure 3.*** *Diagnostic plots showing the convergence characteristics of MCMC chains for five different mixing ratios for the low variance dataset (shown in **Error! Reference source not found.**). Subplots (a) and (b) show variations in the inferred mixing ratio and the error mean with increasing MCMC runs.*

**As it stands, conclusion 1 is favorable and it would indicate that HydroMix is a statistically valid method, but it is not supported by the results. If one replaces the word "uncertainty" by "bias", then the conclusion is supported by the results, but it is not a favorable conclusion anymore and indicates major deficiencies of the used approach. The authors should aim to obtain results that support conclusion 1.**

We do not fully understand the point that the reviewer is trying to make here. Conclusion 1 now reads as:

*"HydroMix gives reliable results for mixing applications with small sample sizes (< 20-30 samples). As expected, the variance in source tracer composition and the ensuing composition overlap determines the uncertainty in the mixing ratio estimates. The uncertainty in mixing ratio estimates increases with increasing variance in source tracer compositions. Mixing ratio estimates improve (in terms of lower error) with increasing number of source samples."*

We believe that the MCMC test along with the model convergence plots prove the reliability and validity of HydroMix.

**Some work is necessary at section 2 (Model description and implementation) to add more clarify and structure to this section. It is hard to understand what the authors are doing here.**
**P4L10 (Section 2): Some serious work is necessary to add more clarify and structure to this section. It is hard to understand what the authors are doing here...**

We moved some of the discussion in the introduction section about linear mixing problems to this section to provide a context behind the theoretical underpinnings of HydroMix (P3L43-P4L4). As also recommended by reviewer 1, we have simplified the explanation in Section 2.1 of the revised manuscript.

**Do you really need so many case studies? They make the paper long and heavy. If you don't need them to make your point, please reduce to 1 or two of them.**
**P7L13-15: Do you really need 5 case studies? They make the paper long and heavy. If you don't need them to make your point, please reduce to 1 or two of them...**

The first case study evaluates whether HydroMix converges to the correct results in standard statistical tests. As also suggested by reviewer 1, we have implemented a MCMC sampler in the benchmark test to prove the validity of the new mixing approach. The second case study evaluates HydroMix using a conceptual hydrologic model and highlights a key deficiency in commonly used mixing approaches. The third case study shows how to account for the deficiency identified earlier. The fourth case study uses HydroMix in a real case study, with the fifth one showing the flexibility offered by this HydroMix to infer additional model parameters. We feel that it is important to demonstrate both the reliability and flexibility of HydroMix and hence have retained the case studies in the revised manuscript.

Responses to the specific comments are mentioned below:

**P2L23-L31, P3L4-12: too methodological for an introduction**

We moved Eq. 1 to the beginning of "Model description and implementation" section in the revised manuscript (P3L43-P4L4). For P3L4-12, we condensed the discussion with key insights from the studies, instead of providing a detailed explanation in the introduction.

**P3L37-38: Some improvement is necessary to provide a proper state-of-the-art with more information on what was exactly done in the many references provided above. Also, here seems to be some missing literature on methods that quantify uncertainty of mixing models. There should be many of them as first studies have been published in the 90s already:**
**Genereux, D., 1998. Quantifying uncertainty in tracer-based hydrograph separations. Water Resour. Res. 34, 915–919. doi:10.1029/98wr00010**

We agree with the reviewer that we had not done an exhaustive review of tracer-based hydrograph separation studies because hydrograph separation is only one of the common applications of mixing models used in hydrology. HydroMix is meant for applications beyond the classic hydrograph separation. However, we do understand that it might be useful to include studies spanning various hydrograph separation techniques and have included a short overview of the same in the revised manuscript (P2L27-L30).

*"In order to express uncertainty in the attribution analysis, a tracer-based hydrograph separation approach was first proposed in the work of Genereux, (1998) and has subsequently been used in many studies (Genereux et al., 2002; Koutsouris and Lyon, 2018; Zhu et al., 2019)."*

**P4L30: The time lag only accounts for advection right? what about dispersion or retardation?**

The time lag accounts for any transport effect, including advection and flow path dispersion. Exchange with tightly bound water (bound to the soil matrix) could lead to retardation effects at much longer time scales than the one considered here. We updated the revised manuscript (P4L23-L25), without, however, adding details of what might potentially cause retardation of stable water isotopes since this would go too far for the context of this paper in our view.

*"In other words, the time lag ($\tau$) stands for any delay caused by tracer transport from the source to the output; we assume that the source components are conservative in nature."*

**P4L42: these studies use the storage selection functions approach, which is somewhat different from the approach introduced here. How do they link to this list?**

The time scale for subsurface flow strongly depends on the catchment moisture conditions. The studies cited (Benettin et al., 2017; Harman, 2015) show that the fraction of young water in streamflow depend on catchment moisture conditions, often referred to as the "inverse storage effect", i.e. there is more event water in the stream when catchment soil moisture is high. That is what we want to say. We have clarified this in the revised manuscript (P4L40-L42):

*"ρ and τ strongly vary in time depending on catchment conditions such as soil moisture (as previously discussed in the context of the 'inverse storage effect' (Benettin et al., 2017; Harman, 2015))."*

**P6L29-40: This is a small state-o-the-art which rather belongs to the introduction. Please keep the methods section as clear as possible.**

This section discusses the different approaches for inferring model parameters, which is why we have retained this in the "*Parameter inference in a Bayesian framework*" section.

**Response to reviewer 3**

We would like to thank this anonymous reviewer for his/her valuable comments and detailed insights into the manuscript. The reviewer raises two major points stated below:

**It remains unclear how the model is initiated and which data coming from field data or other studies were used. In this context, I recommend to report more experimental data, on which you rely in a second step to drive your modelling approach.**
**Figure 3: What do you mean by "different random seeds"? Please see also the general comment regarding the modelling initiation.**

HydroMix is initiated by setting a prior distribution for the model parameters. HydroMix requires concentration data for the different components that mix linearly. In the revised version we have clarified this in P6L43:

*"Apart from the prior distribution of the model parameters, HydroMix requires tracer concentration of the different sources and of the mixture."*

In order to sample the prior distribution of the model parameters, a random number generator is required. To make the results reproducible, it is a recommended practice to specify what is sometimes called the random seed, i.e. a number used to set the random number generator to a specified state, which results in a reproducible set of random numbers (the same seed will give the same random numbers). We have removed this from the text in the revised manuscript to avoid confusion.

As also suggested by the Executive editor, we have provided the experimental data along with the model code through zenodo and included a link of the same in the *"Code and data availability"* section of the revised manuscript.

**The study partly relies on the mixing model of precipitation and snow, not snowmelt. While snow and snowmelt have different isotopic signatures, it is conceptually more correct to use snowmelt for mixing models to estimate its contribution to groundwater. I strongly recommend to better justify why snow instead of snowmelt was used to infer the meltwater supply of groundwater if you cannot extend your calculations to snowmelt.**
**P11L20: It is well known that the isotopic signature of snow is isotopically much more negative and thus much different from the one of snowmelt, which actually forms the 'true liquid' runoff component. Beside, which logistical constraints did prevent from sampling? Snowmelt sampling can easily be carried out by installing PCS collectors or grab sampling the dripping snowmelt**

We agree with the reviewer's point on the usage of snowmelt isotopic ratio instead of snowfall or snowpack isotopic ratio as it is the water from snowmelt that infiltrates and recharges groundwater. However, a recent review of previous investigations that explored the differences in isotopic ratio between snowfall and snowmelt by Beria et al., (2018) revealed that snowfall and snowmelt isotopic ratios have similar mean values but different variances, with lower variance in snowmelt than snowfall isotopic ratios. Snowfall isotopic ratio was found to be the most variable followed by snowpack and then snowmelt isotopic ratio. The lower variance in snowmelt isotopic ratios is because of snowpack homogenization caused by mixing of liquid meltwater between different snowpack layers due to diffusion and dispersion.

We have moved this discussion to Section 4.2 (P15L14-L21):

*"The mixing ratio is estimated with HydroMix using: (1) samples of the isotopic ratio in snowfall, and (2) samples of the isotopic ratio in snowmelt. The two sample distributions differ, as shown in **Error! Reference source not found.**, where the variability of the isotopic ratio is lower in snowmelt when compared to snowfall. In the model at hand, this reduction is obtained because of mixing occurring within the snowpack, leading to homogenization, thus reducing the variability in the isotopic ratio of snowmelt. In field data, such a reduction in variability is also generally observed (Beria et al., 2018), as a result of the homogenization as modelled here and from more complex snow physical processes, which lie beyond the scope of this study."*

There also exists a strong temporal variability in the meltwater isotopic ratios, with more negative isotopic ratios in the early part of the melt season (also referred to as the 'melt-out effect'), and less negative isotopic ratios in the later part of the melt season. However, on an aggregate basis, the mean values of the snowfall and snowmelt isotopic ratios are similar, except in regions with substantial sublimation, which is not the case in the Swiss Alps. This makes snowfall isotopic ratios a reasonable proxy for snowmelt isotopic ratios, which is also supported by results of the synthetic case study (Figure 5). We use snowpack isotopic ratio in the mixing case study because Vallon de Nant is a remotely located catchment with very limited winter access. The catchment experiences frequent winter avalanches and mudslides ("https://www.20min.ch/schweiz/news/story/Schlammlawine-begraebt-Strasse-unter-sich-30186734," 2019), making it hard to monitor during the winter season. This is why we were unable to install snowmelt lysimeters or PCS collectors to regularly sample meltwater. Also, as shown in the review paper of Beria et al., (2018), it is reasonable to replace snowmelt with snowpack isotopic ratios. We have clarified this in P11L25-P12L1:

*"Vallon de Nant is remotely located with very limited winter access, frequently experiencing winter avalanches. Due to these logistical constraints, snowmelt lysimeters or passive capillary samplers could not be setup to sample snowmelt water; accordingly, grab snowpack samples are used here as a proxy for snowmelt."*

**P3L37: Please consider to address also the assumptions of mixing models and the corresponding violation. This aspect may also help to justify Bayesian mixing approaches you describe well.**

Some limitations of the different mixing approaches were already discussed in the original Section 5 (Limitations and Opportunities). We have expanded on the additional assumptions in the Section 5 (P21L10-L31), as also suggested by reviewer 1.

**P4L3: Please add here that the case studies you report later refer to mountainous (high-elevation) catchments**

Thanks for pointing this out. We have mentioned the Vallon de Nant case study here in the revised manuscript (P3L37-L38).

*"The real-world case study applies HydroMix in a high-elevation headwater catchment in Switzerland."*

**P5L17: Please clarify the time-integrated processes you refer to**

This refers to a comment also made by reviewer 1 about the model formulation and the time step change. We have clarified this paragraph in the revised manuscript (P5L14-L19) which now reads as:

*"where the ' signifies the new time-integrated variables. Now, any observed point-scale tracer concentration $p_i$ in a given source $i$ or in the output (e.g., the isotopic ratio of snowmelt) can be assumed to represent a sample from a stationary process (from $S'_1$ or $S'_2$ or $Y'$),. This assumption is in fact implicitly underlying most of the existing hydrological mixing models where point samples are used to characterize a spatial process and where the time reference of the samples is discarded."*

**P8L14-15: How is the sine function defined? How do you derive the amplitude and time lag? Table 4: How do you justify the values of precipitation isotopic lapse rate? Can you provide field data or further references on experimental data?**

The precipitation isotopic ratio time series was assumed to be sampled from a sinusoidal distribution, which is in-line with previous studies in Switzerland (Allen et al., 2018). The mean value and amplitude of the precipitation isotopic ratio closely corresponds with values obtained with the field data in Vallon de Nant. As already pointed out by the Executive editor, we have provided the used database of precipitation isotopic data through zenodo and include the link in the revised manuscript.

We used an offset value of ($-\pi/2$) in the sine function of the precipitation isotopic ratio (mentioned in Table 4). This corresponds to the commonly obtained seasonal variation of the precipitation isotopic ratio in Switzerland, where the ratio is lower (or more negative) during the winters and higher (or less negative) during the summers. Such a seasonal trend has also been reported previously (Allen et al., 2018; Beria et al., 2018).

**P8L22: Please add some references to justify the temperature boundaries you have chosen? Other common boundaries are -1 to 3_C or-1.5 to -1.5, for example.**

We have included the relevant references in the revised manuscript (P8L19).

**P10L39: Please clarify, to which small glaciers do these area proportions of 4.4 % and 10.1 % belong to?**

The line mentioned by the reviewer reads: *"Despite the relatively low elevation, there is a small glacier with an extended moraine that covers 4.4% and 10.1% of the catchment area"*

Vallon de Nant has a small glacier on its South-western tip which covers around 4.4% of the catchment area, below which an extended moraine occupies 10.1% of the catchment area. We have reformulated the text (P10L42-L44) to add clarity:

*"Despite the relatively low elevation, there is a small glacier on its South-western tip, which covers around 4.4% of the catchment area, below which an extended moraine occupies 10.1% of the catchment area."*

**Table 1: What do you mean by top snowpack layer?**

Top snowpack layer refers to the top most layer of the snowpack, which we use as a proxy for recent snowfall as we do not sample snowfall.

**P12L34- 35: Please rephrase and clarify here. The catchment average isotopic ratio does not simply depend on the elevation gradient, which may hold better for precipitation variabilities. But on the presence of the snowpack and where the snowpack is isothermal so that melting could start.**

We agree with the reviewer that the spatial variability in precipitation isotopes is not only a function of elevation, but also depends on the source of moisture origin, cloud condensation temperature and snow metamorphism effects. The case study of lapse rate effect is a mere demonstration that HydroMix allows inference of additional parameters that can account for various physical processes that may modify precipitation isotopic ratio.

With regards to the spatial variability in snowpacks, we use a spatially lumped hydrological model and do not explicitly simulate the spatial variability in snowpack isotopic ratios. We acknowledge that snowpacks at lower elevations melt first and they have isotopic ratios which are different from higher elevation snowpacks. We do not account for this spatial heterogeneity as this lies beyond the scope of this paper. We have expanded upon this in Section 3.5 of the revised manuscript (P13L5-L11):

*"It is important to note that precipitation isotopic ratio is not only a function of elevation, but also depends on other factors such as the source of moisture origin, cloud condensation temperature, secondary evaporation, etc. Similarly, a strong spatial variability exists in the isotopic ratio of snowmelt water, depending on catchment aspect, snow metamorphism, wind distribution, etc. This case study is a mere demonstration that HydroMix allows inference of additional parameters that can account for various physical processes that may modify isotopic ratios."*

**P15L4-8: Please consider moving this paragraph to section 3.2**
The reviewer refers to the following text:
*"The parameters used to generate daily precipitation, air temperature and precipitation isotopic ratios for a run time of 100 years are shown in Table 4. The static volume of groundwater that does not interact directly with the stream, GC is set to 1000 mm. Figure 4 shows the resulting variation in the isotopic ratio of groundwater over the entire 100 year period, showing the system achieves a steady state condition after ~15 years of simulation"*

Thank you for the suggestion. We have moved this to section 3.2 in the revised manuscript.

**Figure 6: How did you define the number of days on which rainfall, snowfall and snowmelt occurred?**

In order to simulate precipitation, time between two successive precipitation events is modelled as a Poisson process, with the number of yearly precipitation events specified as 30. The snow accumulation and the degree-day snowmelt model are then used to compute the number of snowfall days and of snowmelt events. This has been clarified in Section 3.2 (P9L20-L25) in the revised version.

**P17L17, P20L3-59: Please rephrase**

P17L17 reads:
*"In this study, we have used a relatively simple normalization based weighting function (Eq. (24)). Testing other weighting functions which have been proposed in the past (Vasdekis et al., 2014) is certainly possible, and is left for future research."*

The last sentence part is indeed not well formulated. We have rephrase it in the revised manuscript (P17L29-P18L2) as:

*"In this study, we have used a relatively simple normalization based weighting function (Eq. (24)). Testing other weighting functions which have been proposed in the past (Vasdekis et al., 2014) and is left for future research."*

**Figure 9: Please enlarge axis tick labels. Why did you use different x axis scales?**

We have use larger tick labels in the revised manuscript.

The x axis of the top subplot of Figure 9 shows the lapse rate in $^2$H whereas the bottom subplot shows the lapse rate in $^{18}$O. As the isotopic ratios in $^2$H and $^{18}$O are different, the lapse rates are also different, which is why the two subplots have different x-axis scales.

**P22L7: What is small sampling number in your opinion?**

Small sampling sizes can be anywhere less than 20-30 samples. We have mentioned this in the conclusion #1 in revised version (P22L28-29).

**P21L31: I do not see how sediment dynamics fit in here. Sediment dynamics may be coupled with specific runoff components or also decoupled.**

Mixing models are frequently used in sediment fingerprinting to quantify the sediment contribution from different parts of a catchment. Blake et al., (2018) is a recent example where a Bayesian mixing model was used to understand the spatial origin of river sediments. This has been clarified in the revised version (P22L6-L7).

**Typing errors: P1L29: Please change to "that effectively weight"**

**P21L17: Rephrase to "tracer data being available"**

Thanks, this has been rephrased in the revised manuscript.

[revised manuscript text omitted]

---

## Referee Report (RR1)

I appreciate the effort that the authors have made to consider my comments to the previous version of this manuscript. The following issues remain:

**Major comments**

One of my previous comments was about the error bars in the high-variance case for Figure 2 and why they mostly do not cover the true "rho". In the revised version of this manuscript, the error bars have been removed and it is not possible anymore to see if they contain the true mixing ratio. The error bars should be added again, and if the true "rhos" are not contained within, an explanation is needed. If the true "rho" are contained within, then this comment is just a minor one and the issue is resolved. I do acknowledge that limited sample size and overlapping distributions can lead to a bias (i.e. the maximum posterior estimate for "rho" is different from the true one, as visible in Fig. 2a). I do not see, however, why HydroMix would predict that the true mixing ratio is completely impossible (judging from the error bars shown in the previous version of the manuscript). If the amount of information contained in the data is rather low (as in the high-variance case) any consistent Bayesian approach would give rather broad posteriors (since we did not learn much from the data) that would still contain the true value. This deficiency potentially underpins the abiblity of HydroMix to provide reasonable uncertainty estimates of mixing ratios, which is one of its primary goals.

This means that the conclusion on p. 22 line 31: "The uncertainty in mixing ratio estimates increases with increasing variance in source tracer compositions" is not supported by the results. What increases is the bias, not the uncertainty (the uncertainty is the posterior distribution of the mixing ratio, which we cannot see in Figure 2a, we can only see the bias, which is present for high and low mixing ratios).

On p.14 line 7-8 referring to Figure 2b, the authors argue that the error decreases with increasing sample size for the high-variance case. In the caption of Figure 2b, it is written that the presented data is for the low-variance case, however. Which one is true? Is the error also decreasing for the high-variance case?

p.22 line 18-19. "For data scarce environments, this represents an advance over existing probabilistic mixing models that compute mixing ratios […]" This conclusion is not supported by the results. You should either remove that sentence, compare your results to another study that used the existing mixing models which you aim to improve (including a reference to that study), or include such a conventional approach in your synthetic case study and compare the results to your approach. The latter option would be preferable, as it is relatively quickly implemented and does not make the paper much longer in my oppinion. The results could be shown in Figure 2, so no new figure would be needed.

**Minor comments**

Eq. (2): The sources can end up in the target mixture with different time lags. To account for this, Eq. (2) could be written as:

Rho*S1(t-tau1) + (1-rho)*S2(t-tau2)=Y(t)

This could then also be adapted in the text at the corresponding locations and added in the list of difficulties at the bottom of p.4.

A formula for ^Yij is missing (i.e. for the modelled target concentration). I assume it is dependent on the measurements p', not on the process S. In that case, the dependence on S1, S2 has to be removed in Eq. (8,9,10,11, etc.) and replaced by a dependence on p'.

p.5 line 31: sigma is the standard deviation, not the variance. Same on p.5 line 46.

Eq. (9) is wrong. It is correct in the code you made available on github, make sure that the equation in the paper agrees with the code.

p.6 line 23: replace "infer the" by "obtain a sample from the"

p.9 line 13-14: $G_C$ does not avoid zero flow, it avoids zero water in the reservoir. If you reach zero flow or not depends largely how you integrate the differential equations presented in section 3. How do you integrate them from timestep to timestep? This is an important information for any hydrological bucket model. The appendix lists the integration scheme for the isotopic ratios, but not the one for the reservoir levels.

p.9 line 31: "all the rainfall and snowmelt samples": which are they? They have not been mentioned so far.

Eq. (24): can "R" be replaced by $M_S$? Is it the same?

p. 12 line 21: Is it appropriate to cite this study here? Try to give original reference.

Figure2: (a) replace "lambda" by "rho" in figure caption. Replace "Original" by "True" in figure caption. Why where the uncertainty estimates removed from this figure? (b): The sum of the absolute error is a strange measure to use, as it increases with the number of samples. I assume you used the mean error not the sum?

p.20 line 3-5: what do you mean by " this makes the comparison more informative than definitive"? Why is it necessary to have a mechanistic interpretation of a parameter in order to infer that parameter in HydroMix?

p.20 line 11: which uncertainty of the mixing ratio is more realistic, 0.005 or 0.2? If 0.2 is more realistic, then change the value you mentioned in the abstract accordingly.

**Typos**

p.2, line27: "series" instead of "serious"

p.9 line 24: full stop is missing.

---

## Author Response (AR2)

**Submission of revised version of manuscript titled "HydroMix v1.0: a new Bayesian mixing framework for attributing uncertain hydrological sources "**

We would like to thank Bethanna Jackson, David Ham, along with the two reviewers for their valuable feedback. Their inputs led to significant improvements in the manuscript.

We have incorporated the suggestions made by the reviewers in the revised manuscript. As suggested by reviewer #1, we have now included the error bands in figure 2a, which shows that the uncertainty band includes the true mixing ratio. We hope that with these additions and all modifications discussed below, the revised manuscript is now ready for publication in Geoscientific Model Development.

Regards,
Harsh Beria

**Response to reviewer 1**

We would like to thank this anonymous reviewer for his/her second reading of our manuscript and the very useful comments. One main concern raised by the reviewer is about not plotting the uncertainty in the inference of the mixing ratio in the synthetic case study (Figure 2a). We have now included the error bands in figure 2a which shows that the uncertainty band includes the true mixing ratio. We hope this clarifies the reviewer's query. Responses to the specific comments are below. The reviewer comments are in bold, answers in normal style and the quoted text from the revised manuscript in italics.

**One of my previous comments was about the error bars in the high-variance case for Figure 2 and why they mostly do not cover the true "rho". In the revised version of this manuscript, the error bars have been removed and it is not possible anymore to see if they contain the true mixing ratio. The error bars should be added again, and if the true "rhos" _are not contained within, an explanation is needed. If the true "rho" _are contained within, then this comment is just a minor one and the issue is resolved.**

Thanks for pointing this out. We have now added an uncertainty band which shows that the true "rho" is indeed within the range of inferred "rho". The new figure is reproduced below. Details on how the uncertainty band is obtained is included in the caption (see below).

[Figure]

***Figure 1.*** *(a) Scatterplot showing the mixing ratio (ρ) values inferred using HydroMix for the low and high variance synthetic case of **Error! Reference source not found.**. The uncertainty band represents the inferred mixing ratio ± error standard deviation obtained from Eq. 13. The number of source and target samples are 100. (b) Performance of HydroMix in terms of the absolute error between the posterior mixing ratio mean and the true mean for the low*

*variance dataset, summed over all tested ratios plotted as a function of the number of samples drawn for the two sources.*

**This means that the conclusion on p. 22 line 31: "The uncertainty in mixing ratio estimates increases with increasing variance in source tracer compositions" _is not supported by the results. What increases is the bias, not the uncertainty (the uncertainty is the posterior distribution of the mixing ratio, which we cannot see in Figure 2a, we can only see the bias, which is present for high and low mixing ratios).**

Thank you for pointing out this very important distinction. It is indeed the bias of the inferred mixing ratio and not its uncertainty which increases with increasing variance. We have revised the conclusion #1 which now reads as:

*HydroMix gives reliable results for mixing applications with small sample sizes (< 20-30 samples). As expected, the variance in source tracer composition and the ensuing composition overlap determines the bias in the mixing ratio estimates. The bias in mixing ratio estimates increases with increasing variance in source tracer compositions. Mixing ratio estimates improve (in terms of lower error) with increasing number of source samples.*

**On p.14 line 7-8 referring to Figure 2b, the authors argue that the error decreases with increasing sample size for the high-variance case. In the caption of Figure 2b, it is written that the presented data is for the low-variance case, however. Which one is true? Is the error also decreasing for the high-variance case?**

Thank for your pointing this out. We have revised the text as:

*The inferred mean should reproduce the theoretical mean with increasing sample size and we clearly see this for the low variance case in Figure 1. (a) Scatterplot showing the mixing ratio ($\rho$) values inferred using HydroMix for the low and high variance synthetic case of Error! Reference source not found.. The uncertainty band represents the inferred mixing ratio ± error standard deviation obtained from Eq. 13. The number of source and target samples are 100. (b) Performance of HydroMix in terms of the absolute error between the posterior mixing ratio mean and the true mean for the low variance dataset, summed over all tested ratios plotted as a function of the number of samples drawn for the two sources. b, where the model performance markedly improves with increasing number of samples*

We also ran the simulations for the high variance case for which the bias reduce with increasing sample sizes, albeit not for every step change as shown in the figure below.

[Figure]

**P.22 line 18-19. "For data scarce environments, this represents an advance over existing probabilistic mixing models that compute mixing ratios [...]" _This conclusion is not supported by the results. You should either remove that sentence, compare your results to another study that used the existing mixing models which you aim to improve (including a reference to that study), or include such a conventional approach in your synthetic case study and compare the results to your approach. The latter option would be preferable, as it is relatively quickly implemented and does not make the paper much longer in my opinion. The results could be shown in Figure 2, so no new figure would be needed.**

As there are already a lot of case studies in the paper, as was also pointed out by reviewer 2 in the first round of review, we have chosen not to include additional case study. Accordingly, we have revised the text which now reads as:

*This is especially useful in data scarce environments where fitting probability distribution functions is not feasible.*

**Eq. (2): The sources can end up in the target mixture with different time lags. To account for this, Eq. (2) could be written as:**
**Rho*S1(t-tau1) + (1-rho)*S2(t-tau2)=Y(t)**
**This could then also be adapted in the text at the corresponding locations and added in the list of difficulties at the bottom of p.4.**

Thank you for suggesting this, we have revised the text accordingly.

**A formula for Yij is missing (i.e. for the modelled target concentration). I assume it is dependent on the measurements p', not on the process S. In that case, the dependence on S1, S2 has to be removed in Eq. (8,9,10,11, etc.) and replaced by a dependence on p'.**

The formulation for target concentration is stated in Eq. 1 and 2. About the other point on the usage of process vs measurement in the likelihood formulation, we have replaced $S_i$ with $P_i$ in Eq. 8, 9, 10, 11 and 23 in the revised manuscript.

**P5 line 31: sigma is the standard deviation, not the variance. Same on p.5 line 46.**

Thank you for pointing this out. We have revised the text which now reads as:

*Assuming that the residuals can be described with a Gaussian error model with a mean of zero and constant variance $\sigma^2$*

**Eq. (9) is wrong. It is correct in the code you made available on github, make sure that the equation in the paper agrees with the code.**

Thank you for pointing this out, we have rectified the equation which is reproduced below:

$$\log L_j(\tilde{Y}_{obs}|S_1, S_2, \theta) = \sum_{k=1}^{q} \sum_{j=1}^{m} \sum_{i=1}^{n} -0.5 \left[ \log{(2\pi\sigma^2)} + \frac{\left( \tilde{Y}_{obs}^k - \hat{Y}_{ij} \right)^2}{\sigma^2} ) \right].$$

**P.6 line 23: replace "infer the" _by "obtain a sample from the" _**

The line now read as:

*Two methods are traditionally used in hydrology to sample from the posterior distribution from Eq. (11), Markov Chain Monte Carlo (MCMC) sampling (Hastings, 1970; Metropolis and Ulam, 1949) and importance sampling (Glynn and Iglehart, 1989; Neal, 2001)*

**p.9 line 13-14: $G_c$ does not avoid zero flow, it avoids zero water in the reservoir. If you reach zero flow or not depends largely how you integrate the differential equations presented in section 3. How do you integrate them from timestep to timestep? This is an important information for any hydrological bucket model. The appendix lists the integration scheme for the isotopic ratios, but not the one for the reservoir levels.**

Thank you for this important comment. We include $G_c$ in this formulation to avoid zero storage in groundwater reservoir which then might lead to very small outflows. We have specified this in the revised manuscript. The text reads as (P9L13-14):

*… and $G_C$ is a constant groundwater storage that does not interact with the stream (added here to avoid zero storage and thus very small outflow).*

Additionally, we have mentioned the integration scheme used for groundwater storage in the appendix section (Equation 28).

**P.9 line 31: "all the rainfall and snowmelt samples": which are they? They have not been mentioned so far.**

We have mentioned in P9L32-34 that we used the conceptual model to simulate isotopic ratios in rainfall, snowmelt and groundwater. We have now clarified this in the revised manuscript. The new text reads as:

*For the HydroMix application, all the modeled rainfall and snowmelt samples generated using the hydrologic model are used, whereas for groundwater, only one isotopic ratio per month is used (randomly sampled).*

**Eq. (24): can "R" _be replaced by $M_S$? Is it the same?**

For snowmelt, $R$ can be replaced by $M_s$. We used $R_i$ because we wanted to generalize equation (24) for rainfall as well as snowmelt events. To clarify this, we have mentioned the following line:

*where $R_i$ is the snowmelt magnitude and $S_i$ is the isotopic ratio of the $i^{th}$ snowmelt event. Rain weights ($w_j$) are also expressed similarly to Eq. (24).*

**p. 12 line 21: Is it appropriate to cite this study here? Try to give original reference.**

We have now cited the original papers in the revised manuscript.

**Figure2: (a) replace "lambda" _by "rho" _in figure caption. Replace "Original" _by "True" _in figure caption. Why were the uncertainty estimates removed from this figure? (b): The sum of the absolute error is a strange measure to use, as it increases with the number of samples. I assume you used the mean error not the sum?**

We have replaced lambda with mixing ratio as it is more intuitive. We have added an uncertainty band which shows mixing ratio ± error standard deviation. For subplot figure (b), we have computed the median absolute error which is now mentioned in the figure caption and label.

**p.20 line 3-5: what do you mean by " _this makes the comparison more informative than definitive"? Why is it necessary to have a mechanistic interpretation of a parameter in order to infer that parameter in HydroMix?**

In order to avoid ambiguity, we have removed the mechanistic interpretation part of the paragraph. The revised text reads as (P19L17-P20L4):

*The Swiss lapse rate is constructed as a long term spatial average, whereas the inferred isotopic lapse rate in Vallon de Nant is constructed from the temporal variations in the isotopic ratios. These results demonstrate that it is relatively straightforward to jointly infer multiple parameters within the HydroMix modeling framework.*

**P20 line 11: which uncertainty of the mixing ratio is more realistic, 0.005 or 0.2? If 0.2 is more realistic, then change the value you mentioned in the abstract accordingly.**

When we incorporate an additional parameter (isotope lapse rate) during the inference of mixing ratio, we get a wider uncertainty band for the mixing ratio. However, it will not be entirely correct to say that uncertainty in mixing ratio is better estimated with or without incorporating the effect of lapse rate. To avoid ambiguity, we have revised the statement in the abstract which now reads as:

*We then use HydroMix to show that snowmelt accounts for around 61% of groundwater recharge in a Swiss Alpine catchment (Vallon de Nant), despite snowfall only accounting for 40-45% of the annual precipitation.*

**Typos**
**p.2, line27: "series" _instead of "serious" _**
**p.9 line 24: full stop is missing.**
Thank you for pointing these out, we have rectified them in the revised manuscript.

**Response to reviewer 2**

We would like to thank this anonymous reviewer for his/her detailed reading of our manuscript and the response and the additional comments. Responses to the specific comments are below. The reviewer comments are in bold, answers in normal style and the quoted text from the revised manuscript in italics.

**p.2, l.32: variability refers to spatial and temporal variability I assume. Please specify.**

Hydrograph separation is generally used to estimate event vs pre-event water in the stream by using the temporal variability in source tracer concentrations . However, we do understand that hydrograph separation can also be used in a more spatial context. We have clarified this in the revised manuscript (P2L30-L33).

*Bayesian mixing approaches offer a useful alternative to classic hydrograph separation, as Bayesian approaches explicitly acknowledge the temporal variability of source tracer concentrations estimated from observed samples (Barbeta and Peñuelas, 2017; Blake et al., 2018).*

**p.3, l.13: concentrations are "distributed heterogeneously in space", rather than "spatially distributed"**

Thank you for pointing this out. We have revised the text which now reads as:

*An additional complication in hydrology comes from the fact that observed point-scale samples do not necessarily capture the tracer concentrations in the actual sources, which are distributed heterogeneously in space and whose contribution can be temporally variable depending on the state of the catchment (Harman, 2015).*

**p.8, Eq. 15: Is P_s (snow precip) defined previously?**

Snowfall was previously defined in P8L11.

**p.8, l. 31: not explicitly considered, or excluded?**

Rain-on-snow events are not excluded from this study i.e. there can be instances where precipitation happens in the form of liquid rainfall over existing snowpacks and these rainfall events have been included in the study. However, we do not explicitly account for atmospheric heat exchanges happening during rainfall conditions in our snowpack simulation. We have now clarified this in the revised manuscript.

*Enhanced heat exchange processes happening during rain-on-snow events are not explicitly considered as this lies beyond the scope of this paper.*

**p.12, l.21: is this the correct reference? I believe this has been known much longer, e.g. since Dansgaard (1961).**

We have included the original references for 'isotopic lapse rate'.

**Fig 8: do I understand this figure correctly, that d2H and d18O result in different predictions of the fraction of snow in groundwater? Is this because the isotopic composition of the two sources (GW, Snow) is less distinct for one tracer than for the other?**

The prediction of snow fraction are very similar, with large overlap in the uncertainty bands, using d2H and d18O. The slight differences are probably a function of a certain number of fractionated snow samples which do not fall exactly on the local meteoric water line.

**Additional detailed comments:**
The additional comments below were all incorporated in the revised version. Thanks for pointing these imprecisions out.

- **p.1, l.17: you fit a pdf to the source concentrations, not to the sources themselves.**
- **p.3, l.14-17: grammar of the sentence is incorrect. It should be: "…if we were to characterize…., we would need to capture…"**
- **p.4, l.16/17: "tracer concentration" rather than "concentration of tracer" (two instances)**
- **p.7, l.20: I assume "sources" in this case actually refers to the concentrations of these sources, rather than the sources themselves. Please be precise.**
- **p.7, l.44: "stable isotopes of water in deuterium": deuterium IS one of the stable water isotopes, it is not IN the stable water**
- **p.21, l19: what is a "normal model error"? A normally-distributed one? Please use precise terminology.**

[revised manuscript text omitted]